behaviour, ecology

social learning, evolutionary significant units, conservation management, human–wildlife conflict, population viability, cultural transmission

**Authors for correspondence:**
Philippa Brakes
e-mail: p.brakes@exeter.ac.uk
Emma L. Carroll
e-mail: e.carroll@auckland.ac.nz
Ellen C. Garland
e-mail: ecg5@st-andrews.ac.uk

# A deepening understanding of animal culture suggests lessons for conservation

Philippa Brakes[1,2], Emma L. Carroll[3,4], Sasha R. X. Dall[1], Sally A. Keith[5], Peter K. McGregor[6], Sarah L. Mesnick[7,8], Michael J. Noad[9], Luke Rendell[4,10], Martha M. Robbins[11], Christian Rutz[12], Alex Thornton[1], Andrew Whiten[13], Martin J. Whiting[14], Lucy M. Aplin[15,16], Stuart Bearhop[1], Paolo Ciucci[17], Vicki Fishlock[1,18], John K. B. Ford[19], Giuseppe Notarbartolo di Sciara[20], Mark P. Simmonds[21,22], Fernando Spina[23], Paul R. Wade[24,25], Hal Whitehead[26], James Williams[27] and Ellen C. Garland[4,10]

[1]Centre for Ecology and Conservation, Biosciences, College of Life and Environmental Sciences, University of Exeter, Cornwall TR10 9FE, UK
[2]Whale and Dolphin Conservation, Brookfield House, Chippenham, Wiltshire SN15 1LJ, UK
[3]School of Biological Sciences, University of Auckland, Auckland 1010, New Zealand
[4]Sea Mammal Research Unit, School of Biology, University of St Andrews, St Andrews KY16 8LB, UK
[5]Lancaster Environment Centre, Lancaster University, Lancaster LA1 4YQ, UK
[6]ISPA—Instituto Universitário, 1149-041 Lisbon, Portugal
[7]Marine Mammal and Turtle Division, Southwest Fisheries Science Center, National Marine Fisheries Service, NOAA, La Jolla, CA 92037, USA
[8]Scripps Institution of Oceanography, UC San Diego, La Jolla, CA 92093-0203, USA
[9]Cetacean Ecology and Acoustics Laboratory, School of Veterinary Science, The University of Queensland, QLD 4343, Australia
[10]Centre for Social Learning and Cognitive Evolution, School of Biology, University of St Andrews, St Andrews KY16 9TH, UK
[11]Department of Primatology, Max Planck Institute for Evolutionary Anthropology, Leipzig, Germany
[12]Centre for Biological Diversity, School of Biology, University of St Andrews, St Andrews KY16 9TH, UK
[13]Centre for Social Learning and Cognitive Evolution, School of Psychology and Neuroscience, University of St Andrews, St Andrews KY16 9JP, UK
[14]Department of Biological Sciences, Macquarie University, Sydney, NSW 2109, Australia
[15]Max Planck Institute of Animal Behavior, Radolfzell 78315, Germany
[16]Centre for the Advanced Study of Collective Behaviour, University of Konstanz, Konstanz 78467, Germany
[17]Department of Biology and Biotechnologies, University of Rome La Sapienza, 00185 Rome, Italy
[18]Amboseli Trust for Elephants, Langata 00509, Nairobi, Kenya
[19]Department of Zoology, University of British Columbia, Vancouver, BC, Canada
[20]Tethys Research Institute, 20121 Milan, Italy
[21]Bristol Veterinary School, University of Bristol, Bristol BS40 5DU, UK
[22]Humane Society International, London N1 7LY, UK
[23]Istituto Superiore Protezione Ricerca Ambientale (ISPRA), I-40064 Ozzano Emilia (BO), Italy
[24]Marine Mammal Laboratory, Alaska Fisheries Science Center, NOAA Fisheries, Seattle, WA 98115, USA
[25]School of Aquatic and Fishery Sciences, University of Washington, Seattle, WA 98105, USA
[26]Biology Department, Dalhousie University, Halifax, Nova Scotia, Canada B3H4R2
[27]Joint Nature Conservation Committee, Monkstone House, Peterborough PE1 1JY, UK

PB, 0000-0002-2846-1701; ELC, 0000-0003-3193-7288; SRXD, 0000-0001-9873-6507; PKM, 0000-0001-6337-5254; MJN, 0000-0002-2799-8320; LR, 0000-0002-1121-9142; MMR, 0000-0002-6037-7542; CR, 0000-0001-5187-7417; AT, 0000-0002-1607-2047; AW, 0000-0003-2426-5890; MJW, 0000-0002-4662-0227; LMA, 0000-0001-5367-826X; SB, 0000-0002-5864-0129; PC, 0000-0002-0994-3422; VF, 0000-0002-9439-0307; GNdS, 0000-0003-0353-617X; MPS, 0000-0002-3694-843X; PRW, 0000-0003-2428-9323; HW, 0000-0001-5469-3429; ECG, 0000-0002-8240-1267

A key goal of conservation is to protect biodiversity by supporting the long-term persistence of viable, natural populations of wild species. Conservation practice has long been guided by genetic, ecological and demographic indicators of risk. Emerging evidence of animal culture across diverse taxa and its role as a driver of evolutionary diversification, population structure and

demographic processes may be essential for augmenting these conventional conservation approaches and decision-making. Animal culture was the focus of a ground-breaking resolution under the Convention on the Conservation of Migratory Species of Wild Animals (CMS), an international treaty operating under the UN Environment Programme. Here, we synthesize existing evidence to demonstrate how social learning and animal culture interact with processes important to conservation management. Specifically, we explore how social learning might influence population viability and be an important resource in response to anthropogenic change, and provide examples of how it can result in phenotypically distinct units with different, socially learnt behavioural strategies. While identifying culture and social learning can be challenging, indirect identification and parsimonious inferences may be informative. Finally, we identify relevant methodologies and provide a framework for viewing behavioural data through a cultural lens which might provide new insights for conservation management.

## 1. Introduction

A key goal of conservation is to ensure the adaptive potential and long-term persistence of viable populations by maintaining genetic and phenotypic diversity [1]. To achieve this, it is necessary to identify population units in need of conservation, and identify, evaluate and mitigate threats. Standard rubrics for defining units to conserve rely on identifying groups with distinct evolutionary or demographic trajectories (figure 1). International and national conservation frameworks and laws consider the threat status of units to conserve through the assessment of population trajectories, abundance, range dynamics and extinction risk (e.g. IUCN Red List, Endangered Species Act (USA)). We argue that considering animal social learning and animal culture (hereafter 'culture') could augment these conventional conservation approaches and decision-making, by informing the identification of units to conserve and assessing their viability.

The importance of behaviour for conservation biology has been increasingly recognized [2,3]. However, a systematic review of the literature reveals learning and social behaviours were 'rarely considered' in wildlife conservation and management ([4, p. 744]). Our objective is to provide a practical framework to enable conservation managers to consider how culture may impact the viability and structure of certain animal populations and influence animals' responses to conservation strategies. We start by defining animal social learning and culture. We then explore how these processes may influence the transmission of behaviours related to survival and reproduction, and thus provide evidence that social learning might influence demographic processes in a way that impacts population persistence and viability. Next, we delve deeper into the interface of social learning and culture across several behavioural contexts (figure 2). We provide examples where the linkages between conservation and social learning have been demonstrated for endangered species. However, to further elucidate some of the underlying cultural and demographic processes, we also provide examples from species of lower conservation concern, to assist researchers and practitioners in identifying scenarios where social learning may be important for the conservation of endangered species, or for distinct population segments. Finally, we provide a framework (figure 3) to

guide the integration of culture and social learning into current conservation and management efforts for social species.

Acknowledging the bias in the existing literature towards the most studied species, which are often more social and/or viewed as cognitively 'advanced', we highlight the crucial role that cultural transmission can play in guiding effective conservation responses. For example, this was recently achieved through the integration of culture and sociality into aspects of the management framework of the Convention on the Conservation of Migratory Species of Wild Animals (CMS) [5] (electronic supplementary material S1). 'Concerted Actions' approved by the Parties to the treaty, based on cultural data now inform the conservation management of eastern tropical Pacific sperm whales (*Physeter macrocephalus*) and 'nut-cracking' western chimpanzees (*Pan troglodytes verus*) (electronic supplementary material, S1, S4a, S4c) under CMS. Importantly, the aim is not to divert resources from critical conservation needs, or towards cultural species, but to apply scientific knowledge from this field to advance conservation priorities and assist conservation practice.

## 2. Social learning and culture

Social learning has been defined as any learning process that is facilitated by the observation of, or interaction with, another animal or its products [6–8]. An individual may learn new behaviour, like how to open a nut, asocially. Social learning, in contrast, involves the transmission of information from one animal (model) to another (observer), which results in the observer learning the behaviour. Social learning can occur along differing sensory channels (e.g. visual, olfactory) and through a variety of mechanisms such as local enhancement and emulation [8] (electronic supplementary material, S2, glossary). Socially learnt behaviour can flow via: vertical transmission from parent to offspring; oblique transmission from older to younger, often unrelated, individuals; horizontal transmission between peers of the same generation [9]; and even between species [10]. All except the first of these pathways of transmission differ significantly from the dynamics of genetic transmission in the spread of behaviours. It should be noted that, like genetic variation, socially learnt behaviour can be adaptive, non-adaptive or neutral with respect to fitness [11]. However, unlike genetic inheritance, in many circumstances, social learning can facilitate the rapid transmission of behaviour across a diversity of contexts including foraging, migration routes and mate choice [12–16], with potentially significant implications for conservation management.

Social learning may also lead to the transmission of information through groups, giving rise to local behavioural (cultural) variants that persist over time and generations. Culture is defined here as information or behaviours shared within a group and acquired from conspecifics through some form of social learning [7,17]. While this is a broad definition, it allows researchers to identify and measure potential cultural behaviours of conservation value [7]. Culture and its critical foundation, social learning, are observed in a wide variety of different social systems (see [18]). While socially learnt behaviour—and in some cases culture—have increasingly been documented across a wide range of invertebrate and vertebrate species [18], many adaptive behaviours do

(a)

| unit | definition | example in conservation framework |
|---|---|---|
| evolutionary significant unit (ESU) | evolutionary units that show genetic or heritable phenotypic distinctiveness, and that demonstrate isolation, such that there is a restricted flow of information that determines genotype or phenotypes, from other such units [62,63] | IUCN species or sub-species; Canadian Designatable units |
| demographically independent population (DIP) | internal demographic processes (births, deaths) more important to population persistence than migration [60,61] | IUCN subpopulation: distinct groups between which there is little demographic or genetic exchange; US MMPA; IWC populations; Australian EPBC populations |
| cultural variant (CV) | the particular form or variant of the cultural trait (behaviour) displayed by a group or population (derived from [9]) | varies depending on context: can be within, among or equivalent to DIP or ESU. Shown here within DIP (see figure 2 for other examples) |

(b)

**Figure 1.** (a) Description and overview of conservation units (ESUs, DIPs and CVs) and how they are used in current conservation frameworks. (b) Example of the potential relationship between ESUs, DIPs and CVs: one ESU comprises three DIPs of different sizes, with two CVs found at different frequency in each of the DIPs. (Online version in colour.)

not require social input to develop. Conversely, socially learnt behaviour does not necessarily generate sustained or stable cultures, if, for example, it is related to transient resources. Nevertheless, group-wide behavioural variants (or their products) can be assessed to evaluate the possibility that they are socially learnt from conspecifics.

The precautionary principle (electronic supplementary material, S2, glossary) should be applied when assessing the conservation significance of behavioural patterns against the strength of evidence for social learning. For example, in species with endangered populations, information on social learning should rapidly be incorporated into management plans if there is suggestive evidence that these processes might play a role in survival or reproductive rates, even if it is not conclusive [19]. In many species, it is difficult to determine the mechanism of social learning through observation alone. Nevertheless, in a small number of species, including bluehead wrasse (Thalassoma bifasciatum), great tits (Parus major), meerkats (Suricata suricatta), vervet monkeys (Chlorocebus aethiops) and chimpanzees, controlled studies have provided strong evidence that behaviours spread through groups and over generations via social learning [15,16,20–22]. Such work represents a 'gold standard' of evidence for social learning and culture. However, these controlled studies may have ethical implications, or may not be feasible, particularly in the wild or in endangered species, where observed patterns of behavioural expression can instead be used to infer the presence of cultural processes [23–25]. Indeed, controlled studies can be vital for informing conservation by shaping our understanding of the fundamental principles of social learning and cultural transmission, and how they interface with demographic processes (e.g. anti-predator and survival training [26]).

One common tool to detect the presence of culture is the ethnographic method or the method of exclusion, where cultural processes are inferred if ecological and genetic processes can be ruled out [24]. This may reveal a regionally distributed checkerboard of behavioural variants through the examination of multiple populations or social groups spread across the landscape (e.g. [25,27]). However, the exclusion method is vulnerable to both over and under-attribution of cultural causes where researchers fail to recognize subtle environmental factors shaping individual plasticity or genetic

change. For example, chimpanzees' use of long versus short stems to dip for ants was originally thought independent of habitat differences [27], but later detailed studies suggested the choice reflected local variations in the severity of ants' defensive biting [28]. Conversely, the approach may neglect cultural behaviours that are adaptations to different local environments [24], such as tool use to crack shellfish in long-tailed macaques (Macaca fascicularis) [29].

Correlational studies can identify culturally transmitted behaviours where social learning experiments are not possible (e.g. [12]). For example, if the vertical transmission is suspected to play a role in learning foraging strategies, correlations can be assessed between neutral genetic markers, as proxies for relatedness or parental lineages, and stable isotope markers, as proxies for foraging patterns (e.g. [12]). It can be parsimonious to infer that social learning plays a role if a correlation is detected, particularly in species with multiple or generalist foraging strategies which suggest behavioural plasticity or phenotypic variation within a population, or in species where social learning has been previously observed. Vertical culture may be reasonably inferred as a determinant of foraging behaviour, if there is a strong correlation between the foraging measure and a uniparentally inherited genetic marker (e.g. mtDNA) that is unlikely to influence foraging directly [30]. Correlation between functional nuclear DNA markers and foraging behaviour could be indicative of a genetic component to the behaviour, but gene-culture coevolution can also create such patterns [31].

This approach has been questioned in the past due to the assumption that genetics plays a strong role in determining many behaviours [32]. However, the patterns of genetic diversity within populations and species are shaped by the demographic, adaptive and stochastic processes that govern genetic drift, gene flow, mutation and Darwinian selection. In this context, the genetic component of behavioural traits is considered to be shaped by many genes that often have only small effect sizes and moderate heritability [33]. Neutral genetic markers typically used to assess relatedness and parentage are, by definition, less likely to be influenced by Darwinian selection than genes underpinning behavioural variants. While it is sometimes possible to conclusively rule out genetic effects in the described scenario by cross-fostering experiments to discover if they acquire their adopted or

Proc. R. Soc. B 288: 20202718

| behavioural context | species [reference*] | socially learnt behaviour | implications | mitigation strategy | conservation policy |
|---|---|---|---|---|---|
| foraging | killer whales [31] | conservative foraging strategies and vocal dialects | unlikely to switch to alternative prey source when prey preference depleted | managed as distinct cultural units | committee on the status of endangered wildlife in Canada (COSEWIC) DIP, MMPA, ESA |
| | chimpanzees [79] | nut-cracking foraging strategy | potential access to additional food source during dry season when fruit is scarce | investigation of potential benefits or costs of nut-cracking behaviour | CMS concerted action: collaboration across range states |
| | meerkats [20] | social learning and teaching of foraging strategy to young | pups learn to locate and recognise prey, and handle difficult/potentially dangerous prey e.g., scorpions | disturbance that interrupts pathways of transmission may reduce population survival | within population cultural variant |
| | elephants [43] | crop-raiding leads to negative human-wildlife conflict | older males protect and guide younger males; experienced fence breakers serve as repositories of knowledge | 'cultural arms race' requiring ever-adapting management | |
| | griffon vultures [80] | locate food in chain reaction of information transfer | visual contact with a number of conspecifics required to achieve efficient foraging | successful reintroduction: early-stage supplemental food and simultaneous release large numbers individuals in same location | IUCN designated critically endangered |
| | bottlenose dolphins [14] | multiple socially learnt foraging strategies | higher survival rate for tool-using dolphins during heat wave | manage 'sponging', 'shelling' cultural units | within-population cultural variant |
| | golden lion tamarins [26] | knowledge of forage and predator avoidance | reintroduction survival rates extremely low—failed to forage effectively and recognise predators | food provisioning and nest-sites: reintroduced animals survive long enough to socially learn basic life skills | reintroduce with wild elders |
| migration/movement | whooping cranes [67] | socially learnt migratory routes | increased survival resulting from social learning of migration to suitable habitat | reintroduction: use social learning mechanisms to seed migratory routes; social learning important in multiple behavioural contexts | |
| | cod and herring [81] | learning established migratory routes from adults | if ratio juveniles to adults too high oblique transmission interrupted; reduced offspring survival and reproduction | stochastic events required to reintroduce recent released fish to migratory routes | |
| | bighorn sheep [71] | green wave surfing (synching movements with waves of plant growth) | translocated sheep fail to migrate when moved into unfamiliar landscapes | potential to harness emulation or local enhancement by intervening to seed knowledge in a sub-set of individuals | |
| | right whales [12] | socially learnt migratory destinations | unknown flexibility to migration route and destination—high human use areas increase ship strike, noise pollution | vessel restrictions e.g., Boston harbour, for migrating North Atlantic right whales | southern right whale wintering grounds considered DIPs (IWC, New Zealand & Australian domestic legislation), MMPA, ESA |
| | beluga whales [82] | socially learnt migratory routes and destinations; socially learnt avoidance of ice entrapment | exploitation may disrupt beluga societies, impact ability for populations to recover and recolonise habitats where extirpated | maintain viable populations in all habitats, protect migration corridors, continued protection from direct exploitation | Fisheries and Oceans Canada, wildlife management boards, IWC, MMPA, ESA |
| communication | sperm whales [55] | acoustic codas identify populations; provide proxy for foraging strategy; cultural clans have different movement patterns | differential foraging success in different oceanographic conditions; movement patterns important for how clans mitigate effects of El Nino-like events | manage acoustic clans as units to conserve | CMS concerted action: multiple range states collecting data, MMPA, ESA |
| | humpback whales [25] | horizontal transmission of song variants; vocal markers for populations | male sexual display-disruption may impact reproduction but details unclear | rapid assessment tool for identifying and assessing population connectivity; identification of units to conserve | could be incorporated by IWC; MMPA, ESA |
| | New Caledonian crows [83] | vocal dialects | some aspects of tool-assisted foraging behaviour may be socially transmitted, with potential fitness consequences | vocal dialects may provide 'markers' for rapidly mapping variation in tool-related behaviour | |
| | corn buntings [84] | vocal dialects indicate breeding population; lack of clear dialects indicate recent recolonisation, relative unsuitability of habitat | both sexes breed within natal dialect, behaviourally fragmenting continuous population; low settlement density, lack clear dialects indicate habitat suitability | potentially a rapid assessment tool for habitat quality and population connectivity | could be used by national conservation bodies to assess effective population size/habitat suitability |

**Figure 2.** Some examples linking social learning across behavioural contexts, to vital rates and conservation policy. Implications: implications for reproduction, survival or adaptation. Mitigation strategy: mitigation strategy linked to animal culture. *Additional references per species are provided in electronic supplementary material, table S2. Image credits—Chris Huh: humpback whale, killer whale, right whale, sperm whale (https://creativecommons.org/licenses/by-sa/3.0/).

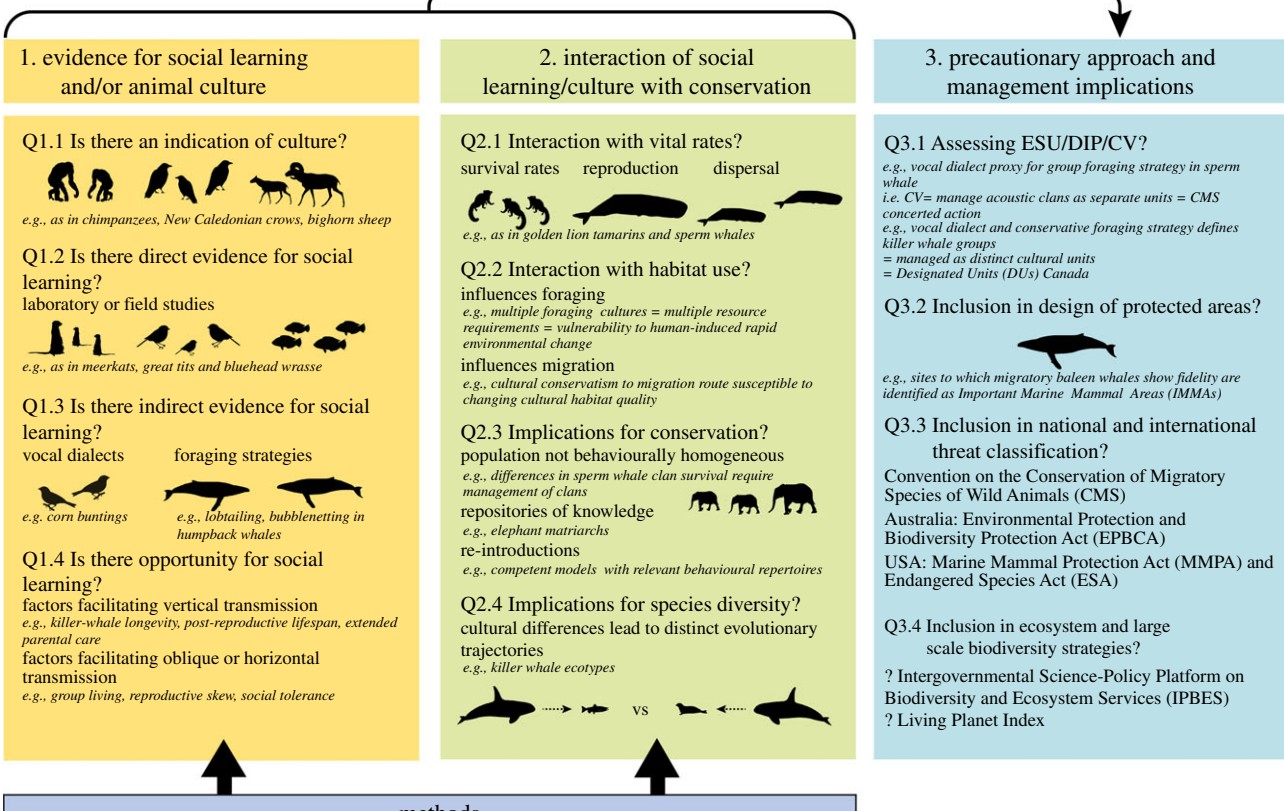

**Figure 3.** A conceptual framework for incorporating evidence and inference on social learning and animal culture into conservation policy and practice (silhouettes indicate examples discussed in main text and electronic supplementary material; see text for details). Image credits—Chris Huh: humpback whale, killer whale, sperm whale; Kent Sorgon: wrasse (https://creativecommons.org/licenses/by-sa/3.0/). (Online version in colour.)

biological parents' foraging strategy [34,35], this is often not ethical or feasible for endangered species.

Culture can be one of many influences that shape behaviour and new modelling approaches now integrate ecological, social and genetic factors into analyses of behavioural variation (e.g. [36]). For example, network-based diffusion analysis (NBDA) has been used to investigate the social transmission of behaviours in chimpanzees [37], humpback whales (*Megaptera novaeangliae* [38]) and bottlenose dolphins (*Tursiops* sp. [39]) by quantifying the extent to which social network structure explains the spread of behaviour [36].

There is no one-size-fits-all method to identify social learning or culture. Feasibility, financial or ethical constraints make it unlikely that some behaviours would ever be definitively shown to be socially learnt. While the inference approaches listed above do not directly test social learning through experiments, they can provide robust, parsimonious inference for the presence of cultural processes underpinned by social learning based on patterns of behavioural expression. Identifying social learning *per se* is important whether or not this social learning gives rise to local cultural variation. Social learning can be a cause, a consequence or a marker of phenotypic diversity, of demography and vital rates, of population genetic structure, and of ecological niche separation (e.g. [40,41]). Conservation outcomes depend on demographic processes. If social learning can influence demography, then it follows that conservation practitioners may benefit from considering cultural processes.

# 3. Conservation through the lens of social learning and culture

Given the conservation challenges associated with rapid environmental change and habitat degradation, maintaining the long-term persistence of viable natural populations requires conservationists to focus on maximizing survival prospects and reproductive outputs of individuals, social groups and populations. To illustrate the links between these demographic parameters and social learning, we draw on examples from a wide variety of species, of the varying threat level. The processes elucidated in these examples have relevance for the management of many species, regardless of their conservation status. Indeed, while some examples in this section may not be of immediate conservation concern, many countries actively manage species and populations to avoid them slipping into such categories; therefore, understanding the influence of culture on demographic processes is highly relevant. Multiple sources of social information can generate the diversity of responses to resource availability and predation pressures [42,43]. What conservation relevant insights might be overlooked by assuming that populations—and social groups— are behaviourally homogeneous? We contend that increasing evidence on social learning and culture provides novel perspectives for addressing this question.

Social learning can create phenotypic variation among individuals and groups that can lead to differences in locating food, developing and propagating specialized foraging

strategies, accessing important habitat or avoiding predators or other risks [18]. Such differences can generate variation in individual fitness within a population and—when such benefits are conferred widely across a social group—can influence vital rates and structure populations [44,45]. First, cultural knowledge may act as a buffer, providing an opportunity to flexibly exploit environments in periods of resource scarcity. Second, in spatially variable environments, social learning can act to 'fine-tune' behaviour to local conditions, a 'resident knowledge' that transient or inexperienced individuals cannot exploit, unless they are able to learn from residents [34]. Third, innovations in response to novel challenges and opportunities can spread via social learning to establish new cultural behaviours, providing a route to exploit new resources [22]. In one of the most famous examples of innovation spread, great and blue (*Cyanistes caeruleus*) tits learnt to break the foil tops of milk bottles delivered to doorsteps and drink the cream beneath, a behaviour that subsequently spread across Britain and Ireland [46]. However, cultural constraints can also limit the spread of adaptive behaviour, depending on the species and context (e.g. [47]).

Quantifying how social learning and culture generate behavioural variation and influence the dynamics of social groups and populations can yield important insights for conservation by examining effects on vital rates. Distilling precisely how social learning and culture can scale up to influence abundance and density, and thus population dynamics, under different scenarios, is challenging. A practical starting point is examining the influence of social learning on two key vital rates—survival and reproduction—as well as the central conservation question of what units to conserve. How population resilience may be impacted is explored in electronic supplementary material, S3.

## (a) Influence of social learning on survival

Building on innovative research on model organisms [20,22,37], consideration and utilization of social learning has proved important for increasing survival in managed populations [2] (electronic supplementary material, S4a). In the case of golden lion tamarins (*Leontopithecus rosalia*), survival rates of reintroduced animals were initially extremely low (13%) [48]. An intensive post-release programme involving supplemental feeding and nest-site provisioning allowed reintroduced animals to survive for long enough to learn basic life skills, doubling survival rates. The offspring of these captive-born re-introduced animals then showed a survival rate of 70%, suggesting that social learning and scaffolding from elders can make a critical contribution to survivorship during reintroductions [26]. In another example, to maximize post-release survival of captive reared critically endangered Hawaiian crows (*Corvus hawaiiensis*), young birds are conditioned to recognize a potential natural predator, the Hawaiian hawk (*Buteo solitarius*), and to exhibit context-appropriate anti-predator behaviour (A. L. Greggor *et al.*, unpublished data). In addition to learning to avoid danger, Hawaiian crows may socially learn key skills required to forage efficiently, communicate in a species-typical manner and breed successfully [49] (see electronic supplementary material, S4a). These examples illustrate the importance of seeking to maintain individuals as 'repositories of knowledge' that may span a number of behavioural contexts and

ensuring individuals scheduled for release are behaviourally competent, thus impacting conservation success.

Social learning can also provide access to novel, high-quality forage, potentially via less energy expenditure than through individual exploration. Socially learnt foraging strategies can also buffer against adverse effects of environmental variability. For example, long-term behavioural studies show bottlenose dolphins in Western Australia have multiple foraging strategies, including socially learnt use of sponges as tools to help extract prey [50]. A recent marine heatwave led to a 5.9% and 12.2% decrease in the survival rate of dolphins that did and did not use tools, respectively. These data indicate that socially transmitted tool use may have buffered a section of the population against the cascading effects of habitat loss on prey species [14]. More broadly, this example highlights how survival in bottlenose dolphins is linked to phenotypic variation. This lesson may be applicable to the conservation and management of other species that show heterogeneity in foraging strategies that could stem from social learning.

## (b) Influence of social learning on reproduction

Variation in reproductive output among females in a population can provide a quantifiable indicator of population health [51] and can be influenced by social learning in complex ways across different scales. For example, individual female bottlenose dolphins in Brazil that specialize in socially learnt cooperative foraging with fishermen may have a fecundity advantage related to increased seasonal prey resources [52]. At a group scale, the sharing of social information by experienced older African elephant (*Loxodonta africana*) matriarchs increases group survival and reproductive success, by providing information on the level of threat posed by elephants from other social groups and by predators in the wider environment [53]. Management plans should incorporate the understanding that matriarchs act as 'repositories of knowledge' and that the loss of these individuals (e.g. culling or translocation) can have population-level impacts that persist for decades [54].

Considering broader population units, sperm whale social units cluster into 'clans' identified by acoustic dialects. Reproductive success varies between clans, which is thought to be associated with socially learnt foraging strategies [7,55] and perhaps alloparental care patterns [56], with potential population-level consequences. Foraging variation among clans can lead sub-populations to respond differently to environmental change, such as the El Niño oceanographic phenomenon. Noting this differential success between acoustic clans, in 2017 the Parties to CMS agreed a Concerted Action to further explore the implications of the clan structure for the conservation of sperm whales in the eastern tropical Pacific [57]. While the influence of social learning on reproductive success is apparent, it is not yet clear how environmental changes influencing feeding success impact clan survival; such information is essential for understanding population dynamics within clans and across the species.

## (c) Influence of social learning and culture on units to conserve

Social learning and culture can promote demographic isolation between groups or populations with relevance to

management and conservation (demographically independent populations (DIPs); figures 1 and 2 [3,47]). For example, killer whales (*Orcinus orca*) can exhibit highly conservative socially learnt prey specializations to the extent that separate, endangered fish-eating Southern Resident killer whale social units forage on fish (e.g. chinook salmon, *Oncorhynchus tshawytscha*) specific to individual river systems [58]. The population abundance of this social unit has declined along with its preferred prey. This reliance on a single river system and cultural reluctance to switch food sources clearly links the importance of understanding foraging culture with conservation management. This demographic isolation can also lead to genetic divergence and speciation through mechanisms such as assortative mating [59]. Figure 2 highlights examples where culture provides valuable data on the delineation of units to conserve at different scales (DIPs [60,61] and evolutionary significant units (ESUs) [59,62]). We direct readers to recent reviews [11,59] that delve into the role of culture as an evolutionary force leading population segments towards distinct evolutionary trajectories as ESUs (figure 1) [41,63] and highlight the role of gene–culture coevolution in this process.

# 4. Ecological studies through the lens of social learning and culture

Evidence for social learning can be identified across several behavioural contexts, perhaps most commonly across the contexts of foraging, migration and communication. These contexts are often the focus of conservation actions. Therefore, our aim is to provide a roadmap to understand the contexts under which social learning may be relevant and to consider ways the field can contribute to promoting conservation outcomes. We hope the examples (electronic supplementary material, S4a–c; figure 2) will encourage readers to re-examine their data using a cultural lens to investigate whether social learning is important for managing and conserving their species.

## (a) Foraging

Social learning plays a vital role in the development of foraging behaviour in many species. Where foraging strategies are socially learnt, innovations can spread rapidly through a social group, facilitating the exploitation of new resources in the environment. For example, young male elephants learn crop-raiding techniques from experienced older males [43] leading to negative conservation outcomes (figure 2). Alternatively, cultural conservatism may lead to an inability to switch prey species despite dwindling resources, as changing foraging techniques to exploit alternative prey may be costly. Failure to recognize that species with multiple foraging cultures may have multiple resource requirements (e.g. killer whales [47]) could undermine conservation efforts.

Direct assessment of diet can be achieved through observations of feeding or using morphological or DNA-based assessments of prey remains found in scat, stomach contents or lavages (e.g. [64]). Stable isotope or fatty acid analyses of tissue or scat can be used to infer foraging location and trophic level [65], where opportunities for direct observations are limited. In one recent example, stable isotope analysis of whisker samples provided strong evidence that young

banded mongooses (*Mungos mungo*) inherit their foraging niche from specific (non-parent) adult cultural role models [35]. Importantly, intraspecific foraging specialization may have real-world consequences for survival and reproduction for endangered species (see electronic supplementary material, S4a). For example, multiple lines of evidence have now established nut-cracking, a foraging specialization limited to sub-populations of critically endangered Western chimpanzees, as a socially learnt and culturally transmitted behaviour that may be essential to survival through the dry season when the fruit is scarce. Noting this specialization and the critically endangered status of these sub-populations, in 2020, the Parties to CMS agreed a Concerted Action to further explore the implications of nut-cracking culture for the conservation of this species (electronic supplementary material, S1 and S4a).

## (b) Migration

In some group-living species or those with extended periods of parental care, the first migration of an individual's life is often with conspecifics. The migration route and/or site learnt can therefore be horizontally transferred from conspecifics [66] or vertically transmitted from parent to offspring (e.g. in whooping cranes, *Grus americana* [67] and southern right whales, *Eubalaena australis* [12]: figure 2), helping ensure that offspring are able to find ephemeral resources in highly patchy environments [68]. Individuals can maintain these socially learnt migratory behaviours across time, leading to a form of cultural conservatism, which can be of relevance to conservation. For example, migratory route fidelity influences management unit designation and the spatially patchy recovery from the hunting of some baleen whale species [40].

Migration movements have been studied directly using field observations and marking methods (e.g. genotypes and photo-identification), and indirectly using stable isotopes and DNA from tissue [12,69]. Genetic pedigrees have been combined with long-term field data, for example, to demonstrate fine-scale extended kin structure at migratory destinations in light-bellied Brent geese (*Branta bernicla hrota*), supporting the hypothesis that site choice has a cultural component [66]. Increasingly, migration movements are studied directly using animal-attached bio-loggers, which provide high-quality fine-scale movement data [70], used to infer links between breeding, stopover and feeding grounds. For example, translocation experiments exploring the cultural basis of migratory behaviour, such as those conducted on big horn sheep (*Ovis canadensis*) and moose (*Alces alces*), provide strong evidence for the importance of cultural behaviour for conservation reintroductions [71] (electronic supplementary material, S4b). Similar patterns are found comparing genetic relatedness and proxies for foraging grounds, such as stable isotopes, in cetacean species (e.g. [12]; figure 2; electronic supplementary material, S4b). Adults with migratory experience and knowledge of suitable habitats may be particularly important as 'knowledgeable individuals' for reintroduction efforts or for preserving existing populations.

## (c) Communication

Vocal communication—the transfer of information or influence between individuals using sound signals—is routinely

studied within the context of social learning and culture using acoustic recordings often supplemented with genetic, identification marks and bio-logging information to provide context (e.g. [72]). Comparisons of vocal differences among groups or populations can require large geographic ranges to be covered, and long-term monitoring for those species that change their vocalizations over time (e.g. via cultural evolution; see electronic supplementary material, S4c). Group-specific or geographic dialect differences become apparent when examining displays across a region and can be used as a cost-effective measure in rapid assessment of population structure [44]. In many cases, cultural conformity to a vocal display within a group appears a key factor in the formation and maintenance of dialects [73]. Acoustic clans in sperm and killer whales offer clear examples of vocal dialects defining groups to conserve, with linkages to vital rates and a CMS Concerted Action in the former, and COSEWIC DIP, USA MMPA and ESA management protection in the latter (figures 2 and 3; electronic supplementary material, S4c). Such vocal differences can be very long lasting and/or lead to reproductive isolation between populations, correlating with genetic differences (e.g. [72,74]). Finally, severe population declines can result in loss of song culture, as shown in critically endangered regent honeyeaters (*Anthochaera phrygia*) [75]; cultural decline may be a precursor to extinction thus providing an important conservation indicator [75].

# 5. Conceptual framework and future directions

Maintaining the adaptive potential and ensuring the long-term persistence of viable natural populations requires conservation managers to focus on maximizing the survival prospects and reproductive outputs of individuals, social groups and populations. An understanding of animal social learning and culture has significant potential to help maximize the impact and efficiency of conservation efforts (electronic supplementary material, table S1). Specifically, understanding linkages between culture and vital rates, cultural evolution, and adaption to rapid global change, will be critical for incorporating culture into management plans. Central to the approach we advocate here is a need to understand the circumstances under which social learning and culture are likely to impact population viability through phenotypic variation (figures 1–3, §3). Additionally, we argue that social learning and culture can be important indicator (§3c) and a resource for resilience in the face of anthropogenic change (figure 2). Social learning and thus cultural evolution may provide opportunities for adaptive behaviours to spread in response to environmental change [76]. Conversely, social learning may prevent the spread of adaptive behaviour, potentially hindering recovery, if conformity is high or some other mechanism promotes cultural 'conservatism' (e.g. killer whale [47]). It may also have a subtle and complex role in resistance to disturbance as the result of knowledgeable elders acting as repositories of social knowledge, as for example in African elephants and killer whales [53,77]. The examples given here are relevant to endangered species, but may also provide insights for those species not currently of conservation concern; managers work to ensure that populations do not decline into threatened status, after all.

Identifying culture and social learning is challenging. While there are a growing number of relatively well-studied species, in the majority of cases, detailed behavioural data are sparse. Indirect identification and parsimonious inferences (e.g. correlation) may therefore be informative. With this perspective in mind, figure 3 provides a framework to guide the integration of data on culture and socially learnt behaviour into current conservation management, and electronic supplementary material, table S1 provides specific recommendations. Within this framework, the first step is to review the evidence, or opportunity, for culture or social learning. Second, how social learning/culture may interact with demographic processes and impact conservation efforts is evaluated and suitable assessment tools are proposed. Third, we suggest how culture could be brought into current conservation frameworks and assessments. For example, if data show that culture or social learning is influencing vital rates of discrete social groups, it could be integrated into population viability analyses. Thus, where salient, phenotypic variation arising from cultural, as well as ecological and genetic processes, could be informative for assessing demographic separation between potential units to manage and conserve [3], and incorporated into national and international conservation frameworks (e.g. IUCN), following published examples (figure 2).

This framework is intended to help guide practitioners towards 'future-proofing' populations by conserving both cultural variation and the capacity for innovation and social learning to maximize the resilience of vulnerable populations. Human activities can both threaten existing cultures and provide a catalyst for new cultural behaviour [13]. The COVID-19 anthropause may provide an opportunity to examine—with an unusual degree of control—the role of social learning in species' responses to significant environmental perturbation [78]. We argue resilience relies on preserving three building blocks of cultural capacity: demography and phenotypic variation; social network structure and population connectivity. Given that such an approach is common to preserving other aspects of biological diversity, and that culture and social learning can interface in multiple ways with conservation efforts, we recommend that the IUCN establish a cross-taxa specialist group to incorporate such information into IUCN assessments. It is only through enhanced collaboration between scientists, conservation practitioners and policy makers that animal culture and social learning can be embedded into conservation practice and policy.

Data accessibility. This article has no additional data.

Authors' contributions. P.B., E.C.G. and E.L.C. co-wrote the manuscript and developed ideas for the figures and tables, with core writing contributions from S.R.X.D., S.A.K., P.K.M., S.L.M., M.J.N., L.R., M.M.R., C.R., A.T., A.W. and M.J.W. All co-authors contributed to the development of ideas and provided feedback on the manuscript.

Competing interests. We declare we have no competing interests.

Funding. Funding and support as follows: Whale and Dolphin Conservation to P.B.; Royal Society New Zealand Rutherford Discovery Fellowship to E.L.C; Radcliffe Fellowship, Radcliffe Institute for Advanced Study, Harvard University, and Biotechnology and Biological Sciences Research Council grant no. [BB/S018484/1]to C.R.; Human Frontier Science Program grant no. [RGP00049] to A.T.; Royal Society University Research Fellowship grant no. [UF160081] to E.C.G.

Acknowledgements. This manuscript is the result of discussions during a 2018 CMS Workshop on Conservation Implications of Animal Culture and Social Complexity, Parma, Italy. We are grateful to the late Bradnee Chambers, former CMS Executive Secretary, for his encouragement and the CMS secretariat for organizing the Parma

workshop; the hosts and sponsors of the workshop (the Appennino Tosco-Emiliano National Park, the Fondazione Monteparma, and the Principality of Monaco); the experts that have supported this initiative; and Vivian Ward for the graphic design of figure 3.

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
