## [Peer Review File · Proceedings of the Royal Society B: Biological Sciences]

Review History

RSPB-2020-2718.R0 (Original submission)

Review form: Reviewer 1

Recommendation

Major revision is needed (please make suggestions in comments)

Scientific importance: Is the manuscript an original and important contribution to its field?
Excellent

General interest: Is the paper of sufficient general interest?
Good

Quality of the paper: Is the overall quality of the paper suitable?
Excellent

Is the length of the paper justified?
Yes

Should the paper be seen by a specialist statistical reviewer?

No

Do you have any concerns about statistical analyses in this paper? If so, please specify them explicitly in your report.

No

It is a condition of publication that authors make their supporting data, code and materials available - either as supplementary material or hosted in an external repository. Please rate, if applicable, the supporting data on the following criteria.

Is it accessible?

N/A

Is it clear?

N/A

Is it adequate?

N/A

Do you have any ethical concerns with this paper?

No

Comments to the Author

See attached file. (See Appendix A)

Review form: Reviewer 2

Recommendation

Accept with minor revision (please list in comments)

Scientific importance: Is the manuscript an original and important contribution to its field?

Good

General interest: Is the paper of sufficient general interest?

Excellent

Quality of the paper: Is the overall quality of the paper suitable?

Good

Is the length of the paper justified?

Yes

Should the paper be seen by a specialist statistical reviewer?

No

Do you have any concerns about statistical analyses in this paper? If so, please specify them explicitly in your report.

No

It is a condition of publication that authors make their supporting data, code and materials available - either as supplementary material or hosted in an external repository. Please rate, if applicable, the supporting data on the following criteria.

Is it accessible?

N/A

Is it clear?

N/A

Is it adequate?

N/A

Do you have any ethical concerns with this paper?

No

Comments to the Author

This manuscript covers an interesting and important topic – the significance of understanding and harnessing animal culture to improve conservation outcomes. In general, the authors nicely articulate the multitude of ways that exploiting knowledge of social learning and cultural transmission of behavior can be used to bolster conservation capacity. I believe that this manuscript will be of interest to a broad readership, including both behavioral ecologists and conservation biologists. It is in my opinion, however, that several topics (e.g., limitations of the ethnographic approach, gene-culture association, cultural evolution, phylogenetic approaches) lacked depth or were partially or vaguely described. Further, and in attempt to adhere to journal word/page limits (I assume), the authors relegated many citations for empirical evidence needed support their messaging to the supplemental material. I focused my review on constructive ways to strengthen the manuscript and I hope the authors find my comments helpful.

Major comments:

Ln 111-113: A central promise of the manuscript is “to provide recommendations (table S1) and a framework (figure 3) to guide the integration of culture and social learning into current conservation efforts”. For this reason, I suggest moving table S1 into the main manuscript (it could feature as a full-page table). I understand that space is limited, but if the editor is willing to accommodate space for this table, I think it will strengthen the manuscript. The table would benefit from some horizontal lines that delineate and organize the various topics/ideas. As currently constructed, the table is difficult to follow.

Ln 181-190: This paragraph is important and could be strengthened using an example (or two) from the literature to help non-experts solidly understand the potential consequences of applying the ethnographic approach. If an example of the potential ‘pitfalls’ of this approach are absent in the literature, offering a hypothetical example would be helpful.

Ln 192-207: This is an important discussion that, to my understanding, is not well resolved in the literature. Using neutral genetic markers (i.e., those not under selection) to demonstrate correlation or clustering of behavioral variants among related individuals does indeed suggest inheritance of said behavior. Nevertheless, whether such inheritance of behavior has a genetic or socially learned basis cannot be determined via a correlational approach because genes that exert control of behavior may tag along with neutral genetic markers. Thus, correlation alone does not allow researchers to disentangle a genetic versus a socially learned basis of behavior (Laland and Janik 2006). Despite substantial effort, our ability to identify genes that control specific aspects of behavior are extremely limited (e.g., and with regards to migration, see Franchini et al. 2017 and references within), meaning determining the mode of inheritance usually requires an experiment. To deal with this issue, previous authors suggest transplant experiments as a viable approach for teasing apart genetic versus learned aspects of behavior (e.g., Laland and Janik 2006). As currently written, it seems the authors are suggesting that any evidence of inheritance of behavior can be viewed as evidence of social learning. For this reason, I suggest adding further discussion regarding the unresolved and complex associations between genotypic data and behavior.

Laland, K. N., and V. M. Janik. 2006. The animal cultures debate. *Trends in Ecology & Evolution* 21:542-547.

Franchini, P., I. Irisarri, A. Fudickar, A. Schmidt, A. Meyer, M. Wikelski, and J. Partecke. 2017. Animal tracking meets migration genomics: Transcriptomic analysis of a partially migratory bird species. *Molecular Ecology* 26:3204-3216.

Ln 380: Suggest expanding this section a bit and discussing gene-culture evolution (Whitehead et al. 2019, *Nature Comm.*, already cited in manuscript) in more detail than is currently presented in the manuscript.

Ln 401: Cultural conservatism of migration routes is not mentioned in main text but is featured in figure 3. Suggest a sentence or two on conservatism. Also, such conservatism is generally referred to "site fidelity" (e.g., see Switzer 1993), and in the migration literature as "route fidelity" (e.g., see Morrison et al. in press, Wyckoff et al. 2018, Berger et al. 2006, Sawyer and Kauffman 2011)

Switzer, P. V. 1993. Site fidelity in predictable and unpredictable habitats. *Evolutionary Ecology* 7:533-555.

Wyckoff, T. B., H. Sawyer, S. E. Albeke, S. L. Garman, and M. J. Kauffman. 2018. Evaluating the influence of energy and residential development on the migratory behavior of mule deer. *Ecosphere* 9:e02113.

Berger, J., Cain, S.L. & Berger, K.M. (2006) Connecting the dots: an invariant migration corridor links the Holocene to the present. *Biology Letters*, 22, 528–531.

Sawyer, H. & Kauffman, M.J. (2011) Stopover ecology of a migratory ungulate. *Journal of Animal Ecology*, 80, 1078-1087.

Morrison, Thomas, J. Merkle, Grant Hopcraft, E. O. Aikens, J. L. Beck, R. B. Boone, A. Courtemanch et al. Cross-species comparison of drivers of site fidelity in ungulates. *Journal of Animal Ecology* (2020).

<https://eprints.gla.ac.uk/226574/>

Ln 403-405: Suggest adding a hypothetical or literature-based example here. Otherwise, reads as an unsupported assertion (which it is not).

Ln 424: No mention of the use of translocation experiments for understanding cultural basis of foraging and migratory behavior (or any other behavior) in the migratory or other sections of the manuscript. Again, see Laland and Janik 2006. Also see Jesmer et al. 2018, which is used as an example in figure 3 but is not cited in the manuscript.

Jesmer, B. R., J. A. Merkle, J. R. Goheen, E. O. Aikens, J. L. Beck, A. B. Courtemanch, M. A. Hurley, D. E. McWhirter, H. M. Miyasaki, K. L. Monteith, and M. J. Kauffman. 2018. Is ungulate migration culturally transmitted? Evidence of social learning from translocated animals. *Science* 361:1023-1025.

Ln 501: Here and in figure 3, the authors need to provide more information about how assessing phylogenetic propensity for social learning/cultural transmission would be accomplished. Even if it is just a sentence or two with some citations. Otherwise, the proposed approach is vague and requires the reader to guess and make leaps of faith as to what the authors are suggesting.

Ln 518-521: Here and other areas of the manuscript, particularly the section on resilience, resistance, and recovery, is an opportune place to integrate some discussion of the role of cultural evolution in conservation outcomes. For example, cultural evolution may be a source of rapid

adaptation to various aspects of global change.

Figure 2: In the case of bighorn sheep in North America, the mitigation strategy suggested by the authors is not viable. In Jesmer et al. 2018, bighorn sheep and moose were translocated into areas of their historic range where the species was previously extirpated. So, there were no knowledgeable individuals to translocate because no individuals had experience on those landscapes. I do not see a way to introduce/translocate knowledgeable individuals in reintroduction efforts. One potential mitigation strategy would be to perform some sort of 'assisted migration' (not in the way the term is typically used) where humans harness emulation or local enhancement to 'seed' knowledge in a subset of individuals (perhaps by training captive reared individuals in the natural setting a la Mueller et al. 2013, Science) about where and when to migrate, then see/hope if that information diffuses through a population.

Although space is always limited in print journals, and authors often feel forced to shift as much information as possible to supplemental materials (myself included), I strongly suggest a 7th column of references be added here. The information in the supplement will relatively rarely be read and authors of these works will not get appropriate accreditation. Given most of the studies used in this table are cited in the main text, I do not think it will add too many citations to the author's reference section to add a references column.

Figure 3: In figure 3 and throughout the manuscript I suggest spelling out acronyms (e.g., ESU, DIP, CV, CMS, CITES, IPBES, COSEWIC, etc.). These acronyms are not used over and over throughout the manuscript, so spelling them out will not cost much space but will certainly make the manuscript easier to read for non-experts. Also, with regards to using acronyms in figure 3, non-expert readers will have to search through the main text or supplemental materials to decipher the acronyms and interpret the figure.

Also, a more minor point regarding figure 3 is that bighorn lambs do not have full-curl horns. Suggest finding a silhouette with no or very small horns or using image editing software to delete the horns on the current silhouette.

Minor comments:

Ln 163: What are "precautionary principles"? If this is jargon, I suggest defining. If the authors are referring to discussion preceding this paragraph, it remains unclear as to what these principles are.

Ln 175-179: This is an important message that I think will be appreciated by those interested in studying animal culture in wild populations where such 'gold standards' are impractical or impossible to implement. Thank you.

Ln 209: Suggest describing what is meant by "viewing culture in isolation" or reword. Unclear what is meant here.

Ln 219: Two complete sentences combined with a comma (i.e., run on sentence).

Ln 389: Typo. Should read "may not yet be"

Ln 389-393: Reads awkwardly, suggest moving "thus providing a fertile area for on-going research" to the end of the sentence.

Ln 438-441: Suggest moving this sentence to after the sentence on the use of genetic pedigrees. Otherwise, the sentence on biologging interrupts the logical linkage between the first and third sentences about the use of genotypic data.

Decision letter (RSPB-2020-2718.R0)

21-Dec-2020

Dear Dr Brakes:

Your manuscript has now been peer reviewed and their comments (not including confidential comments to the Editor) are included at the end of this email for your reference. As you will see, the reviewers have raised a fair number of concerns with your manuscript but, overall, they were both positive about the goals and overall pitch of the review. So, I would like to invite you to revise your manuscript to address their concerns.

Research ethics:

Use of animals and field studies:

It is a condition of publication that you make available the data and research materials supporting the results in the article (<https://royalsociety.org/journals/authors/author-guidelines/#data>). Datasets should be deposited in an appropriate publicly available repository and details of the associated accession number, link or DOI to the datasets must be included in the Data Accessibility section of the article (<https://royalsociety.org/journals/ethics->

policies/data-sharing-mining/). Reference(s) to datasets should also be included in the reference list of the article with DOIs (where available).

Please submit a copy of your revised paper, ideally within three weeks. If we do not hear from you within this time your manuscript will be rejected. If you are unable to meet this deadline please let us know as soon as possible, as we may be able to grant a short extension. We do understand that this is Christmas and we all deserve a holiday!

Best wishes, and have a very happy Christmas,

Innes Cuthill
Prof. Innes Cuthill
Reviews Editor, Proceedings B
mailto: proceedingsb@royalsociety.org

Reviewer(s)' Comments to Author:

Referee: 1

Comments to the Author(s)

See attached file.

Referee: 2

Comments to the Author(s)

This manuscript covers an interesting and important topic – the significance of understanding and harnessing animal culture to improve conservation outcomes. In general, the authors nicely articulate the multitude of ways that exploiting knowledge of social learning and cultural

transmission of behavior can be used to bolster conservation capacity. I believe that this manuscript will be of interest to a broad readership, including both behavioral ecologists and conservation biologists. It is in my opinion, however, that several topics (e.g., limitations of the ethnographic approach, gene-culture association, cultural evolution, phylogenetic approaches) lacked depth or were partially or vaguely described. Further, and in attempt to adhere to journal word/page limits (I assume), the authors relegated many citations for empirical evidence needed support their messaging to the supplemental material. I focused my review on constructive ways to strengthen the manuscript and I hope the authors find my comments helpful.

Major comments:

Ln 111-113: A central promise of the manuscript is “to provide recommendations (table S1) and a framework (figure 3) to guide the integration of culture and social learning into current conservation efforts”. For this reason, I suggest moving table S1 into the main manuscript (it could feature as a full-page table). I understand that space is limited, but if the editor is willing to accommodate space for this table, I think it will strengthen the manuscript. The table would benefit from some horizontal lines that delineate and organize the various topics/ideas. As currently constructed, the table is difficult to follow.

Ln 181-190: This paragraph is important and could be strengthened using an example (or two) from the literature to help non-experts solidly understand the potential consequences of applying the ethnographic approach. If an example of the potential ‘pitfalls’ of this approach are absent in the literature, offering a hypothetical example would be helpful.

Ln 192-207: This is an important discussion that, to my understanding, is not well resolved in the literature. Using neutral genetic markers (i.e., those not under selection) to demonstrate correlation or clustering of behavioral variants among related individuals does indeed suggest inheritance of said behavior. Nevertheless, whether such inheritance of behavior has a genetic or socially learned basis cannot be determined via a correlational approach because genes that exert control of behavior may tag along with neutral genetic markers. Thus, correlation alone does not allow researchers to disentangle a genetic versus a socially learned basis of behavior (Laland and Janik 2006). Despite substantial effort, our ability to identify genes that control specific aspects of behavior are extremely limited (e.g., and with regards to migration, see Franchini et al. 2017 and references within), meaning determining the mode of inheritance usually requires an experiment. To deal with this issue, previous authors suggest transplant experiments as a viable approach for teasing apart genetic versus learned aspects of behavior (e.g., Laland and Janik 2006). As currently written, it seems the authors are suggesting that any evidence of inheritance of behavior can be viewed as evidence of social learning. For this reason, I suggest adding further discussion regarding the unresolved and complex associations between genotypic data and behavior.

Laland, K. N., and V. M. Janik. 2006. The animal cultures debate. *Trends in Ecology & Evolution* 21:542-547.

Franchini, P., I. Irisarri, A. Fudickar, A. Schmidt, A. Meyer, M. Wikelski, and J. Partecke. 2017. Animal tracking meets migration genomics: Transcriptomic analysis of a partially migratory bird species. *Molecular Ecology* 26:3204-3216.

Ln 380: Suggest expanding this section a bit and discussing gene-culture evolution (Whitehead et al. 2019, *Nature Comm.*, already cited in manuscript) in more detail than is currently presented in the manuscript.

Ln 401: Cultural conservatism of migration routes is not mentioned in main text but is featured in figure 3. Suggest a sentence or two on conservatism. Also, such conservatism is generally referred to “site fidelity” (e.g., see Switzer 1993), and in the migration literature as “route fidelity” (e.g., see Morrison et al. in press, Wyckoff et al. 2018, Berger et al. 2006, Sawyer and Kauffman 2011)

Switzer, P. V. 1993. Site fidelity in predictable and unpredictable habitats. *Evolutionary Ecology* 7:533-555.

Wyckoff, T. B., H. Sawyer, S. E. Albeke, S. L. Garman, and M. J. Kauffman. 2018. Evaluating the influence of energy and residential development on the migratory behavior of mule deer. *Ecosphere* 9:e02113.

Berger, J., Cain, S.L. & Berger, K.M. (2006) Connecting the dots: an invariant migration corridor links the Holocene to the present. *Biology Letters*, 22, 528–531.

Sawyer, H. & Kauffman, M.J. (2011) Stopover ecology of a migratory ungulate. *Journal of Animal Ecology*, 80, 1078–1087.

Morrison, Thomas, J. Merkle, Grant Hopcraft, E. O. Aikens, J. L. Beck, R. B. Boone, A. Courtemanch et al. Cross-species comparison of drivers of site fidelity in ungulates. *Journal of Animal Ecology* (2020).
<https://eprints.gla.ac.uk/226574/>

Ln 403-405: Suggest adding a hypothetical or literature-based example here. Otherwise, reads as an unsupported assertion (which it is not).

Ln 424: No mention of the use of translocation experiments for understanding cultural basis of foraging and migratory behavior (or any other behavior) in the migratory or other sections of the manuscript. Again, see Laland and Janik 2006. Also see Jesmer et al. 2018, which is used as an example in figure 3 but is not cited in the manuscript.

Jesmer, B. R., J. A. Merkle, J. R. Goheen, E. O. Aikens, J. L. Beck, A. B. Courtemanch, M. A. Hurley, D. E. McWhirter, H. M. Miyasaki, K. L. Monteith, and M. J. Kauffman. 2018. Is ungulate migration culturally transmitted? Evidence of social learning from translocated animals. *Science* 361:1023-1025.

Ln 501: Here and in figure 3, the authors need to provide more information about how assessing phylogenetic propensity for social learning/cultural transmission would be accomplished. Even if it is just a sentence or two with some citations. Otherwise, the proposed approach is vague and requires the reader to guess and make leaps of faith as to what the authors are suggesting.

Ln 518-521: Here and other areas of the manuscript, particularly the section on resilience, resistance, and recovery, is an opportune place to integrate some discussion of the role of cultural evolution in conservation outcomes. For example, cultural evolution may be a source of rapid adaptation to various aspects of global change.

Figure 2: In the case of bighorn sheep in North America, the mitigation strategy suggested by the authors is not viable. In Jesmer et al. 2018, bighorn sheep and moose were translocated into areas of their historic range where the species was previously extirpated. So, there were no knowledgeable individuals to translocate because no individuals had experience on those landscapes. I do not see a way to introduce/translocate knowledgeable individuals in reintroduction efforts. One potential mitigation strategy would be to perform some sort of 'assisted migration' (not in the way the term is typically used) where humans harness emulation or local enhancement to 'seed' knowledge in a subset of individuals (perhaps by training captive reared individuals in the natural setting a la Mueller et al. 2013, *Science*) about where and when to migrate, then see/hope if that information diffuses through a population.

Although space is always limited in print journals, and authors often feel forced to shift as much information as possible to supplemental materials (myself included), I strongly suggest a 7th column of references be added here. The information in the supplement will relatively rarely be read and authors of these works will not get appropriate accreditation. Given most of the studies

used in this table are cited in the main text, I do not think it will add too many citations to the author's reference section to add a references column.

Figure 3: In figure 3 and throughout the manuscript I suggest spelling out acronyms (e.g., ESU, DIP, CV, CMS, CITES, IPBES, COSEWIC, etc.). These acronyms are not used over and over throughout the manuscript, so spelling them out will not cost much space but will certainly make the manuscript easier to read for non-experts. Also, with regards to using acronyms in figure 3, non-expert readers will have to search through the main text or supplemental materials to decipher the acronyms and interpret the figure.

Also, a more minor point regarding figure 3 is that bighorn lambs do not have full-curl horns. Suggest finding a silhouette with no or very small horns or using image editing software to delete the horns on the current silhouette.

Minor comments:

Ln 163: What are "precautionary principles"? If this is jargon, I suggest defining. If the authors are referring to discussion preceding this paragraph, it remains unclear as to what these principles are.

Ln 175-179: This is an important message that I think will be appreciated by those interested in studying animal culture in wild populations where such 'gold standards' are impractical or impossible to implement. Thank you.

Ln 209: Suggest describing what is meant by "viewing culture in isolation" or reword. Unclear what is meant here.

Ln 219: Two complete sentences combined with a comma (i.e., run on sentence).

Ln 389: Typo. Should read "may not yet be"

Ln 389-393: Reads awkwardly, suggest moving "thus providing a fertile area for on-going research" to the end of the sentence.

Ln 438-441: Suggest moving this sentence to after the sentence on the use of genetic pedigrees. Otherwise, the sentence on biologging interrupts the logical linkage between the first and third sentences about the use of genotypic data.

Author's Response to Decision Letter for (RSPB-2020-2718.R0)

See Appendix B.

RSPB-2020-2718.R1 (Revision)

Review form: Reviewer 1

Recommendation

Reject – article is not of sufficient interest (we will consider a transfer to another journal)

Scientific importance: Is the manuscript an original and important contribution to its field?

Good

General interest: Is the paper of sufficient general interest?

Good

Quality of the paper: Is the overall quality of the paper suitable?

Good

Is the length of the paper justified?

Yes

Should the paper be seen by a specialist statistical reviewer?

No

Do you have any concerns about statistical analyses in this paper? If so, please specify them explicitly in your report.

No

It is a condition of publication that authors make their supporting data, code and materials available - either as supplementary material or hosted in an external repository. Please rate, if applicable, the supporting data on the following criteria.

Is it accessible?

N/A

Is it clear?

N/A

Is it adequate?

N/A

Do you have any ethical concerns with this paper?

No

Comments to the Author

This is the second time I have reviewed "A deepening understanding of animal culture brings novel perspectives to conservation". The authors did a good job addressing most of my previous comments and I thank them for taking the time to do so. However, I still can't help but get the feeling that this paper primarily about animal social learning and culture and the conservation aspect is only considered secondarily. Perhaps I am just not getting it – but I still don't feel like the link to conservation is very strong and I still found that many of the examples are great in the context of culture and social learning, but they are less relevant for the conservation and management of species. For example, that culture is related to survival in bottlenose dolphins (a very common species) is not particularly relevant for the conservation of any other species.

Within section 3 'Conservation through the lens of social learning and culture' – I found the link to conservation to be tenuous. The introductory section provides examples are only vaguely linked to potential conservation issues through things like "resource scarcity" – an ecological process that happens in all environments for all species (so this is not special for species at risk). See below for comments on the specific line numbers.

As this section progresses through section (a) to section (b) I felt as though I was reading a review on the link between culture and demography, not culture and conservation. Conservation certainly includes aspects of demography and the role of culture and social learning on vital rates is very important, but I still feel like the link to conservation is not as central as it could be. Specifically, I think the implication here is that if culture or social learning influence survival or

reproduction than this is good for conservation. Sure, this may be true, but I feel

I do think the golden lion tamarin example is a good one, but I find the bottlenose dolphin example to be a bit more tangential – aren't bottlenose dolphins one of the most common dolphin species in the world? I think the dots need to be better connected from social learning -> survival -> conservation efforts (in the tamarin example, the authors do this, but I think the dolphin example on line 297-301 one falls short).

Similarly in section (b), I find the bottlenose dolphin example between lines 321-324 is really interesting and important – but what is the link to conservation? Same thing goes for the sperm whale example between lines 329-333 – what is the link to conservation?

One general comment I have is that in many places, conservation is focused on habitat protection and less focused on protecting a single species – but this does not really come up except only indirectly in the migration section. What is the relevance of culture or social learning within the context of protecting large swathes of habitats?

Perhaps the authors might consider shifting the focus of the paper away from the extensive series of examples provided in text and using this space to focus more generally on the link between culture/social learning and the variable of interest (i.e. survival, reproduction, units to conserve, foraging, migration, or communication). I think having the examples in Figure 2 is great, but so many of the in text examples don't have a connection to conservation. I think getting more into the ecological and evolutionary underpinnings of culture and social learning and linking these general ideas to the conservation and management of species or habitats might be a more compelling way to demonstrate the importance of culture/social learning. With this type of narrative the authors could then develop more of a theoretical framework that does not rely on so many disparate empirical examples and they could focus on providing more detail on a smaller number of examples (e.g. the golden lion tamarin example).

I hope my review has been helpful and apologize if it is disappointing or if I am wrong in my assessment.

Decision letter (RSPB-2020-2718.R1)

03-Mar-2021

Dear Dr Brakes:

This is a tricky one -- I asked you to revise your manuscript because one referee was extremely positive and the criticisms of the other referee looked as though they could be addressed without too much difficulty. However, having sent the revision back to the latter referee, they have come back not impressed... and now I see the nub of their problem. If I'd picked up on this earlier, I would have rejected the manuscript first time round, but now I feel I have led you too far down an encouraging route to deliver that slap in the face. The problem is that the title "A deepening understanding of animal culture brings novel perspectives to conservation" does not describe the paper. As the referee says, below, there is a very good review of social learning and animal culture, and some suggestions about how this MIGHT bring new perspectives to conservation, but it is definitely not a review of HOW deepening understanding of animal culture brings novel perspectives to conservation. As the referee says, the case studies are either not about animals of conservation concern or, when they are, the impacts are often speculative. The referee does recommend 'reject' on this basis, but also suggests that the ms be saved by reframing it (not least with a change of title) as a review of our understanding of animal culture and how this might have lessons for conservation. I think that message is an important one, which is why I am

giving you a second chance to revise and resubmit. I stress that this is a reframing issue, not any problem with examples you use or the arguments you put forward.

When submitting your revision please upload a file under "Response to Referees" in the "File Upload" section. This should document, point by point, how you have responded to the reviewer's and Editors' comments, and the adjustments you have made to the manuscript. We require a copy of the manuscript with revisions made since the previous version marked as 'tracked changes' to be included in the 'response to referees' document.

Research ethics:

Use of animals and field studies:

It is a condition of publication that you make available the data and research materials supporting the results in the article (<https://royalsociety.org/journals/authors/author-guidelines/#data>). Datasets should be deposited in an appropriate publicly available repository and details of the associated accession number, link or DOI to the datasets must be included in the Data Accessibility section of the article (<https://royalsociety.org/journals/ethics-policies/data-sharing-mining/>). Reference(s) to datasets should also be included in the reference list of the article with DOIs (where available).

Please submit a copy of your revised paper within three weeks. If we do not hear from you within this time your manuscript will be rejected. If you are unable to meet this deadline please let us know as soon as possible, as we may be able to grant a short extension.

Best wishes,
Innes

Prof Innes Cuthill
Reviews Editor, Proceedings B
mailto: proceedingsb@royalsociety.org

Reviewer(s)' Comments to Author:

Referee: 1

Comments to the Author(s)

This is the second time I have reviewed "A deepening understanding of animal culture brings novel perspectives to conservation". The authors did a good job addressing most of my previous comments and I thank them for taking the time to do so. However, I still can't help but get the feeling that this paper primarily about animal social learning and culture and the conservation aspect is only considered secondarily. Perhaps I am just not getting it – but I still don't feel like the link to conservation is very strong and I still found that many of the examples are great in the context of culture and social learning, but they are less relevant for the conservation and management of species. For example, that culture is related to survival in bottlenose dolphins (a very common species) is not particularly relevant for the conservation of any other species.

Within section 3 'Conservation through the lens of social learning and culture' – I found the link to conservation to be tenuous. The introductory section provides examples are only vaguely linked to potential conservation issues through things like "resource scarcity" – an ecological process that happens in all environments for all species (so this is not special for species at risk). See below for comments on the specific line numbers.

As this section progresses through section (a) to section (b) I felt as though I was reading a review on the link between culture and demography, not culture and conservation. Conservation certainly includes aspects of demography and the role of culture and social learning on vital rates is very important, but I still feel like the link to conservation is not as central as it could be. Specifically, I think the implication here is that if culture or social learning influence survival or reproduction than this is good for conservation. Sure, this may be true, but I feel

I do think the golden lion tamarin example is a good one, but I find the bottlenose dolphin example to be a bit more tangential – aren't bottlenose dolphins one of the most common dolphin species in the world? I think the dots need to be better connected from social learning -> survival

-> conservation efforts (in the tamarin example, the authors do this, but I think the dolphin example on line 297-301 one falls short).

Similarly in section (b), I find the bottlenose dolphin example between lines 321-324 is really interesting and important – but what is the link to conservation? Same thing goes for the sperm whale example between lines 329-333 – what is the link to conservation?

One general comment I have is that in many places, conservation is focused on habitat protection and less focused on protecting a single species – but this does not really come up except only indirectly in the migration section. What is the relevance of culture or social learning within the context of protecting large swathes of habitats?

Perhaps the authors might consider shifting the focus of the paper away from the extensive series of examples provided in text and using this space to focus more generally on the link between culture/social learning and the variable of interest (i.e. survival, reproduction, units to conserve, foraging, migration, or communication). I think having the examples in Figure 2 is great, but so many of the in text examples don't have a connection to conservation. I think getting more into the ecological and evolutionary underpinnings of culture and social learning and linking these general ideas to the conservation and management of species or habitats might be a more compelling way to demonstrate the importance of culture/social learning. With this type of narrative the authors could then develop more of a theoretical framework that does not rely on so many disparate empirical examples and they could focus on providing more detail on a smaller number of examples (e.g. the golden lion tamarin example).

I hope my review has been helpful and apologize if it is disappointing or if I am wrong in my assessment.

Author's Response to Decision Letter for (RSPB-2020-2718.R1)

See Appendix B.

Decision letter (RSPB-2020-2718.R2)

24-Mar-2021

Dear Ms Brakes

I am pleased to inform you that your manuscript entitled "A deepening understanding of animal culture suggests lessons for conservation" has been accepted for publication in Proceedings B.

If you are likely to be away from e-mail contact during this period, let us know. Due to rapid publication and an extremely tight schedule, if comments are not received, we may publish the paper as it stands.

Data Accessibility section

Open access

You are invited to opt for open access via our author pays publishing model. Payment of open access fees will enable your article to be made freely available via the Royal Society website as soon as it is ready for publication. For more information about open access publishing please visit our website at http://royalsocietypublishing.org/site/authors/open_access.xhtml.

The open access fee is £1,700 per article (plus VAT for authors within the EU). If you wish to opt for open access then please let us know as soon as possible.

Paper charges

Sincerely,

Proceedings B

Appendix A

This manuscript is call to action of sorts for the inclusion of culture and social learning within the field of conservation biology. The manuscript is well written, comprehensive, and undoubtedly fills an important gap in the literature. I would agree with the authors in a broad sense that culture, social learning, and more broadly social behaviour are extremely important in our goal to conserve species at risk – especially highly social species. I have a number of broad comments that I think authors might consider either adding or emphasizing a little bit to ensure the manuscript is accessible to a broader audience.

Social species vs. non-social species: The article is framed around the idea that culture and social learning are important to consider for conservation – which I very much agree with – but as is the case for most conservation issues, incorporating culture and social learning into conservation is not a one size fits all solution. For example, many of the examples in text as well as those provided in Table 1 and supplementary section S3 tend to be species that are highly social, cognitively ‘advanced’, and tend to live in fission-fusion societies. I don’t necessarily think this bias is a fault of the manuscript – I think these are simply the species most studied in the context of culture/social learning.

But, it does beg the question: is culture/social learning only applicable when discussing the conservation of highly social/cognitively advanced/fission-fusion species? I would say not, but the authors might consider adding a caveat at some point (possibly around lines 111-113?) that this framework and the associated recommendations are most relevant for social species.

At line 113 they authors could say “to guide the integration of culture and social learning into current conservation efforts of highly social species” – or something to that affect. Another option would be to additional sections in the supplementary section S3 that detail some species that are not highly social and cognitively advanced, perhaps there are some examples from fish, invertebrates, or small mammals? If not, then I think the main text deserves a short section outlining the existing bias in the literature.

What about conservation behaviour? I found there is somewhat of a glaring omission of any discussion on the topic of conservation behaviour. Numerous systematic papers have come out in the last year that discuss conservation behaviour in a similar context to what the authors do in this manuscript. Specifically, Berger-Tal et al. 2016 Conservation Biology examines the number of papers that examine different axes of behaviour and conservation. Although they do not include culture in their analysis they do include learning (as well as communication, migration, and foraging – all behaviours the authors suggest can have cultural underpinnings). Based on figure 1 from Berger-Tal et al. 2016, it looks like there are some studies that link learning to various conservation threats and solutions – something the authors discuss throughout the manuscript – but no mention is made of the past efforts of those tackling the same problem through the lens of conservation behaviour. I’d suggest the authors consider adding a sub-section (possibly in section 3) of the manuscript that outlines discusses and integrates the existing work that exists in the conservation behaviour literature.

Why should managers care? I ask this is somewhat of a rhetorical question. I couldn't agree more with the authors that culture, social learning, and social behaviour more generally are exceptionally important in the conservation and management of species at risk. However, in my own experience studying species at risk and communicating with managers, it is not necessarily something that scientists are great at communicating to managers and managers might not always be receptive to the idea that these things are important. Further, I think it is unlikely that any (or many) managers are going to be reading this article, so the onus then falls onto the researcher studying culture/social learning to pass along this information to managers they work with through regular briefing meetings or through research grants/proposals/reports. So, that being said, I would challenge the authors to consider providing advice or recommendations (possibly in a box or figure?) that provides guidance for researchers to communicate with managers. For example, if your student studying culture/social learning of an endangered species was meeting with a government or NGO conservation manager, what are 2-3 ways you would encourage them to justify the importance of culture (keeping in mind things like communicating in plain language and with concision/precision).

I realize much of this information is contained within the article and in Table S1, but I think it would be useful for students and others new to working directly with conservation managers to have a generic and accessible summary/rubric that they could follow.

Examples: there are a ton of examples throughout the paper, which is great! However, it isn't always clear whether the example is meant to simply be an example of an example that culture/social learning exists in a given context or whether it is meant to be an example of how culture/social learning was important in that context for managing/conserving an endangered species. E.g. Lines 399-400 – this example of elephant crop raiding is certainly an example of social learning and foraging, but it doesn't really have anything to do with the conservation of elephants. In fact, it's almost the opposite... elephants learning bad behaviour at the risk of being shot! I think for most of the examples in the main text it might be a bit more precise if the authors could make sure it is clear whether a given example is meant to simply demonstrate that a given ecological trait (e.g. foraging, migration, communication) is related to culture/social learning OR whether these things are related AND it had an impact on the conservation of that species. Other examples on lines 412-415 (banded mongoose), 431 (whooping cranes), line 432 (right whales), line 443 (Brent geese), line 463 (chickens), 474 (white crowned sparrows). I realize the details for some of these species are outlined further in the supplementary materials, but I think when possible, it is good to clarify in text exactly what the example is doing for the reader, e.g. is it a culture ~ foraging relationship or is it a culture ~ foraging relationship that had an impact on conservation.

Minor comments:

Section 3(d): this section is quite short (only 3 sentences). By contrast, section 3(c) is two paragraphs. I would suggest either expanding on the ideas here or removing it altogether – I am not sure 3 sentences does the topic justice.

Figure 2: maybe the authors could add a column that has the IUCN red list listing for these species? Again, I think this comes back to my earlier comment about how it isn't always clear whether an example is simply an example of a relationship between culture and another variable or whether this relationship is relevant for conservation. I think providing information on the species listing will add clarity to this aspect of the manuscript.

Figure 3: are the silhouettes under each question indicative that those species satisfy the requirements associated with the question? E.g. Q1.1 Is there an indication of culture? The silhouettes of a primate, bird, and sheep – does this mean these species have satisfied the requirement of culture? And does that necessarily mean that any species silhouettes under Q1.2 or Q1.3 have not met the requirement for culture? Please provide a more detailed caption specifically regarding the placement of silhouettes.

Appendix B

27th January 2021

Re-submission of review article: 'A deepening understanding of animal culture brings novel perspectives to conservation'

Dear Dr Kren and Prof Cuthill,

thank you for providing an extension for responding to the Reviewer's comments for our manuscript 'A deepening understanding of animal culture brings novel perspectives to conservation'. We thank the Reviewers for their support and also the constructive comments to improve, clarify and streamline our manuscript. We have endeavored to address all points raised within the length limitations of the manuscript. Line numbers provided here refer to the revised manuscript without track changes.

Reviewer 1

This manuscript is call to action of sorts for the inclusion of culture and social learning within the field of conservation biology. The manuscript is well written, comprehensive, and undoubtedly fills an important gap in the literature. I would agree with the authors in a broad sense that culture, social learning, and more broadly social behaviour are extremely important in our goal to conserve species at risk – especially highly social species. I have a number of broad comments that I think authors might consider either adding or emphasizing a little bit to ensure the manuscript is accessible to a broader audience.

Social species vs. non-social species: The article is framed around the idea that culture and social learning are important to consider for conservation – which I very much agree with – but as is the case for most conservation issues, incorporating culture and social learning into conservation is not a one size fits all solution. For example, many of the examples in text as well as those provided in Table 1 and supplementary section S3 tend to be species that are highly social, cognitively 'advanced', and tend to live in fission-fusion societies. I don't necessarily think this bias is a fault of the manuscript – I think these are simply the species most studied in the context of culture/social learning.

But, it does beg the question: is culture/social learning only applicable when discussing the conservation of highly social/cognitively advanced/fission-fusion species? I would say not, but the authors might consider adding a caveat at some point (possibly around lines 111-113?) that this framework and the associated recommendations are most relevant for social species.

At line 113 they authors could say "to guide the integration of culture and social learning into current conservation efforts of highly social species" – or something to that affect. Another option would be to additional sections in the supplementary section S3 that detail some species that are not highly social and cognitively advanced, perhaps there are some examples from fish, invertebrates, or small mammals? If not, then I think the main text deserves a short section outlining the existing bias in the literature.

A: We agree with the reviewer that incorporating culture and social learning into conservation is not a one size fits all solution. The species included in the text and tables are those for which the implications for conservation have been investigated and provide the best evidence. Noting this, we have endeavoured to mitigate bias by selecting as wide a variety of species examples as possible. In providing these examples, we hope this manuscript will serve as a 'call to arms' for research on species that are not highly social/cognitively advanced/fission-fusion species.

We have incorporated the suggested wording into the manuscript L113 "Finally, we provide a framework (figure 3) to guide the integration of culture and social learning into current conservation efforts for social species".

We further explicitly acknowledge the bias in the literature and have added the following text L116 "Acknowledging the bias in the existing literature towards the most studied species, which are often more social and/or viewed as cognitively 'advanced', we highlight the crucial role that cultural transmission can play in guiding effective conservation responses".

What about conservation behaviour? I found there is somewhat of a glaring omission of any discussion on the topic of conservation behaviour. Numerous systematic papers have come out in the last year that discuss conservation behaviour in a similar context to what the authors do in this manuscript. Specifically, Berger-Tal et al. 2016 Conservation Biology examines the number of papers that examine different axes of behaviour and conservation. Although they do not include culture in their analysis they do include learning (as well as communication, migration, and foraging – all behaviours the authors suggest can have cultural underpinnings). Based on figure 1 from Berger-Tal et al. 2016, it looks like there are some studies that link learning to various conservation threats and solutions – something the authors discuss throughout the manuscript – but no mention is made of the past efforts of those tackling the same problem through the lens of conservation behaviour. I'd suggest the authors consider adding a sub-section (possibly in section 3) of the manuscript that outlines discusses and integrates the existing work that exists in the conservation behaviour literature.

A: We have edited the text to incorporate this important point. A subsection is not possible due to space limitations; instead, we have made this point explicit in the Introduction to alert the reader to this complementary literature. We thank the reviewer for the suggested reference to include. L101 "The importance of conservation behaviour has been increasingly recognized [2,3]. However, a systematic review of the literature reveals learning and social behaviours were 'rarely considered' in wildlife conservation and management [5 p.744]".

Why should managers care? I ask this is somewhat of a rhetorical question. I couldn't agree more with the authors that culture, social learning, and social behaviour more generally are exceptionally important in the conservation and management of species at risk. However, in my own experience studying species at risk and communicating with managers, it is not necessarily something that scientists are great at communicating to managers and managers might not always be receptive to the idea that these things are important. Further, I think it is unlikely that any (or many) managers are going to be reading this article, so the onus then falls onto the researcher studying culture/social learning to pass along this information to managers they work with through regular briefing

meetings or through research grants/proposals/reports. So, that being said, I would challenge the authors to consider providing advice or recommendations (possibly in a box or figure?) that provides guidance for researchers to communicate with managers. For example, if your student studying culture/social learning of an endangered species was meeting with a government or NGO conservation manager, what are 2-3 ways you would encourage them to justify the importance of culture (keeping in mind things like communicating in plain language and with concision/precision).

I realize much of this information is contained within the article and in Table S1, but I think it would be useful for students and others new to working directly with conservation managers to have a generic and accessible summary/rubric that they could follow.

A: We have edited Table S1 to provide headings to assist in clarity and highlight some important points a student or those new to the field could convey. We have added text to highlight two key points in the main manuscript L493 “Specifically, understanding linkages between culture and vitals rates, cultural evolution, and adaption to rapid global change, will be critical for incorporating culture into management plans”.

Examples: there are a ton of examples throughout the paper, which is great! However, it isn't always clear whether the example is meant to simply be an example of an example that culture/social learning exists in a given context or whether it is meant to be an example of how culture/social learning was important in that context for managing/conserving an endangered species. E.g. Lines 399-400 – this example of elephant crop raiding is certainly an example of social learning and foraging, but it doesn't really have anything to do with the conservation of elephants. In fact, it's almost the opposite... elephants learning bad behaviour at the risk of being shot! I think for most of the examples in the main text it might be a bit more precise if the authors could make sure it is clear whether a given example is meant to simply demonstrate that a given ecological trait (e.g. foraging, migration, communication) is related to culture/social learning OR whether these things are related AND it had an impact on the conservation of that species. Other examples on lines 412-415 (banded mongoose), 431 (whooping cranes), line 432 (right whales), line 443 (Brent geese), line 463 (chickens), 474 (white crowned sparrows). I realize the details for some of these species are outlined further in the supplementary materials, but I think when possible, it is good to clarify in text exactly what the example is doing for the reader, e.g. is it a culture ~ foraging relationship or is it a culture ~ foraging relationship that had an impact on conservation.

A: This is a very good point. Where there is a clear conservation/management impact, we have highlighted this in the text to help with clarity [see changes made at L316, L333, L336, L350, L380-381, L415-419, L430-433, L467].

Minor comments:

Section 3(d): this section is quite short (only 3 sentences). By contrast, section 3(c) is two paragraphs. I would suggest either expanding on the ideas here or removing it altogether – I am not sure 3 sentences does the topic justice.

A: In order to further elucidate Section 3(d), we moved section 3(c) to the electronic supplementary material (ESM, S3) and added the following text beginning at L353: “Social learning and culture can promote demographic isolation between groups or populations with

relevance to management and conservation (demographically independent populations (DIPs); figure 1, [48,61]). This demographic isolation can lead to genetic divergence and speciation through mechanisms such as assortative mating[62]. Figure 2 highlights examples where culture provides valuable data on the delineation of units to conserve at different scales (DIPs ([63,64]) and ESUs ([62,65]; figure 2). We direct readers to recent reviews [11,62] that delve into the role of culture as an evolutionary force leading population segments towards distinct evolutionary trajectories as ESUs (figure 1, [41,66]), and highlight the role of gene-culture co-evolution in this process”.

Figure 2: maybe the authors could add a column that has the IUCN red list listing for these species? Again, I think this comes back to my earlier comment about how it isn't always clear whether an example is simply an example of a relationship between culture and another variable or whether this relationship is relevant for conservation. I think providing information on the species listing will add clarity to this aspect of the manuscript.

A: Thank you for this suggestion. We explored including the IUCN listing for the species examples represented in Figure 2. However, our feeling is that many of the IUCN global listing – where they do not include higher resolution assessments of regional populations - are too coarse to represent the finer-scale processes we are highlighting. For example, humpback whales are listed globally by the IUCN as of 'least concern' (LC), whereas the Oceania humpback whale population is listed as endangered (E). We articulate this issue in the text beginning at L510, where we state: “Thus, where salient, phenotypic variation arising from cultural, as well as ecological and genetic processes, could be informative for assessing demographic separation between potential units to conserve [61] and incorporated into national and international conservation frameworks (e.g., IUCN), following published examples (figure 2)”. A key recommendation on this point is provided in the penultimate sentence (L526), which states: “Given that such an approach is common to preserving other aspects of biological diversity, and that culture and social learning can interface in multiple ways with conservation efforts, we recommend that the IUCN establish a cross-taxa specialist group to incorporate such information into IUCN assessments”.

Figure 3: are the silhouettes under each question indicative that those species satisfy the requirements associated with the question? E.g. Q1.1 Is there an indication of culture? The silhouettes of a primate, bird, and sheep – does this mean these species have satisfied the requirement of culture? And does that necessarily mean that any species silhouettes under Q1.2 or Q1.3 have not met the requirement for culture? Please provide a more detailed caption specifically regarding the placement of silhouettes.

A: To address this point we added further detail to the caption for Figure 3, noting that the placement of the silhouettes refer back to examples discussed in the main text (Figure 3 caption, L879) and for additional clarity placed the common names next to the silhouettes in the Figure. The new caption reads thus: “A conceptual framework for incorporating evidence and inference on social learning and animal culture into conservation policy and practice to mitigate human-induced rapid environmental change (HIREC), (silhouettes indicate examples discussed in main text and supplement; for explanatory detail, see text).

Reviewer 2

Comments to the Author(s)

This manuscript covers an interesting and important topic— the significance of understanding and harnessing animal culture to improve conservation outcomes. In general, the authors nicely articulate the multitude of ways that exploiting knowledge of social learning and cultural transmission of behavior can be used to bolster conservation capacity. I believe that this manuscript will be of interest to a broad readership, including both behavioral ecologists and conservation biologists. It is in my opinion, however, that several topics (e.g., limitations of the ethnographic approach, gene-culture association, cultural evolution, phylogenetic approaches) lacked depth or were partially or vaguely described. Further, and in attempt to adhere to journal word/page limits (I assume), the authors relegated many citations for empirical evidence needed support their messaging to the supplemental material. I focused my review on constructive ways to strengthen the manuscript and I hope the authors find my comments helpful.

Major comments:

Ln 111-113: A central promise of the manuscript is “to provide recommendations (table S1) and a framework (figure 3) to guide the integration of culture and social learning into current conservation efforts”. For this reason, I suggest moving table S1 into the main manuscript (it could feature as a full-page table). I understand that space is limited, but if the editor is willing to accommodate space for this table, I think it will strengthen the manuscript. The table would benefit from some horizontal lines that delineate and organize the various topics/ideas. As currently constructed, the table is difficult to follow.

A: We thank the reviewer for this suggestion. After discussions regarding space limitations with the editor, we now emphasise the framework (Fig. 3) which summarises the recommendations in table S1. We have further improved table S1 by adding horizontal lines and subheadings, as requested, following both Reviewer’s suggestions for improvements.

Ln 181-190: This paragraph is important and could be strengthened using an example (or two) from the literature to help non-experts solidly understand the potential consequences of applying the ethnographic approach. If an example of the potential ‘pitfalls’ of this approach are absent in the literature, offering a hypothetical example would be helpful.

*A: We have included two examples, as requested, to help non-experts understand the potential consequences of applying the ethnographic approach. L188 “However, the exclusion method is vulnerable to both over and under-attribution of cultural causes where researchers fail to recognise subtle environmental factors shaping individual plasticity or genetic change. For example, chimpanzees’ use of long versus short stems to dip for ants was originally thought independent of habitat differences [26], but later detailed studies suggested the choice reflected local variations in the severity of ants’ defensive biting [27]. Conversely the approach may neglect cultural behaviours that are adaptations to different local environments [24], such as tool use to crack shellfish in long-tailed macaques (*Macaca fascicularis*) [28].”*

Ln 192-207: This is an important discussion that, to my understanding, is not well resolved in the literature. Using neutral genetic markers (i.e., those not under selection) to demonstrate correlation or clustering of behavioral variants among related individuals does indeed suggest inheritance of said behavior. Nevertheless, whether such inheritance of behavior has a genetic or socially learned basis cannot be determined via a correlational approach because genes that exert control of behavior may tag along with neutral genetic markers. Thus, correlation alone does not allow researchers to disentangle a genetic versus a socially learned basis of behavior (Laland and Janik 2006). Despite substantial effort, our ability to identify genes that control specific aspects of behavior are extremely limited (e.g., and with regards to migration, see Franchini et al. 2017 and references within), meaning determining the mode of inheritance usually requires an experiment. To deal with this issue, previous authors suggest transplant experiments as a viable approach for teasing apart genetic versus learned aspects of behavior (e.g., Laland and Janik 2006). As currently written, it seems the authors are suggesting that any evidence of inheritance of behavior can be viewed as evidence of social learning. For this reason, I suggest adding further discussion regarding the unresolved and complex associations between genotypic data and behavior.

Laland, K. N., and V. M. Janik. 2006. The animal cultures debate. *Trends in Ecology & Evolution* 21:542-547.

Franchini, P., I. Irisarri, A. Fudickar, A. Schmidt, A. Meyer, M. Wikelski, and J. Partecke. 2017. Animal tracking meets migration genomics: Transcriptomic analysis of a partially migratory bird species. *Molecular Ecology* 26:3204-3216.

Ln 380: Suggest expanding this section a bit and discussing gene-culture evolution (Whitehead et al. 2019, *Nature Comm.*, already cited in manuscript) in more detail than is currently presented in the manuscript.

A: To expand upon some of these important issues, we have included the following new paragraph to help clarify the above discussion points (L215-237). However, as we cannot accommodate all requests for expansion of text, we explicitly direct readers to two excellent recent reviews discussing gene culture evolution which also cover the topic of section 3(c):

“This approach has been questioned in the past due to the assumption that genetics plays a strong role in determining many behaviours [32]. However, the patterns of genetic diversity within populations and species are shaped by the demographic, adaptive and stochastic processes that govern genetic drift, gene flow, mutation and Darwinian selection. In this context, behavioural traits are likely determined by many genes that often have only small effect sizes and moderate heritability [33]. Neutral genetic markers typically used to assess relatedness and parentage are, by definition, less likely to be influenced by Darwinian selection than genes underpinning behavioural variants. While it is sometimes possible to conclusively rule out genetic effects in the described scenario by cross-fostering experiments to discover if they acquire their adopted or biological parents’ foraging strategy [34,35], this is often not ethical or feasible for endangered species.

Culture can be one of many influences that shape behaviour and new modelling approaches now integrate ecological, social and genetic factors into analyses of behavioural variation (e.g., [36]). For example, network-based diffusion analysis (NBDA) has been used to

investigate the social transmission of behaviours in chimpanzees [37], humpback whales (Megaptera novaeangliae [38]), and bottlenose dolphins (Tursiops sp. [39]) by quantifying the extent to which social network structure explains the spread of a behaviour [36].”

Ln 401: Cultural conservatism of migration routes is not mentioned in main text but is featured in figure 3. Suggest a sentence or two on conservatism. Also, such conservatism is generally referred to “site fidelity” (e.g., see Switzer 1993), and in the migration literature as “route fidelity” (e.g., see Morrison et al. in press, Wyckoff et al. 2018, Berger et al. 2006, Sawyer and Kauffman 2011)

Switzer, P. V. 1993. Site fidelity in predictable and unpredictable habitats. *Evolutionary Ecology* 7:533-555.

Wyckoff, T. B., H. Sawyer, S. E. Albeke, S. L. Garman, and M. J. Kauffman. 2018. Evaluating the influence of energy and residential development on the migratory behavior of mule deer. *Ecosphere* 9:e02113.

Berger, J., Cain, S.L. & Berger, K.M. (2006) Connecting the dots: an invariant migration corridor links the Holocene to the present. *Biology Letters*, 22, 528–531.

Sawyer, H. & Kauffman, M.J. (2011) Stopover ecology of a migratory ungulate. *Journal of Animal Ecology*, 80, 1078–1087.

Morrison, Thomas, J. Merkle, Grant Hopcraft, E. O. Aikens, J. L. Beck, R. B. Boone, A. Courtemanch et al. Cross-species comparison of drivers of site fidelity in ungulates. *Journal of Animal Ecology* (2020).

<https://eprints.gla.ac.uk/226574/>

A: We have included the following edits to address this point (L408) and have brought information from the ESM into the main body of the manuscript. We also now specifically mention route and/or site fidelity in this section:

*“In contrast, in some group-living species or those with extended periods of parental care, the first migration of an individual’s life is often with conspecifics. The migration route and/or site learnt can therefore be horizontally transferred from conspecifics [71] or vertically transmitted from parent to offspring (e.g., in whooping cranes, *Grus americana* [72] and southern right whales, *Eubalaena australis* [29]: see figure 2) helping ensure that offspring are able to find ephemeral resources in highly patchy environments [73]. Individuals can maintain these socially learned migratory behaviours across time, leading to a form of cultural conservatism, which can be of relevance to conservation. For example, migratory route fidelity influences management unit designation and the spatially patchy recovery from hunting of some baleen whale species [40].”*

Ln 403-405: Suggest adding a hypothetical or literature-based example here. Otherwise, reads as an unsupported assertion (which it is not).

A: We have been more explicit here by noting the killer whale example (in Whitehead 2010) in L385.

Ln 424: No mention of the use of translocation experiments for understanding cultural basis of foraging and migratory behavior (or any other behavior) in the migratory or other sections of the manuscript. Again, see Laland and Janik 2006. Also see Jesmer et al. 2018, which is used as an example in figure 3 but is not cited in the manuscript.

Jesmer, B. R., J. A. Merkle, J. R. Goheen, E. O. Aikens, J. L. Beck, A. B. Courtemanch, M. A. Hurley, D. E. McWhirter, H. M. Miyasaki, K. L. Monteith, and M. J. Kauffman. 2018. Is ungulate migration culturally transmitted? Evidence of social learning from translocated animals. *Science* 361:1023-1025.

*A: This information is included in the expanded genetics section and as a Section in the ESM (S4b). We have directed readers towards evidence on translocation experiments provided by Jesmer et al. 2018 in the main text, by adding the following sentence L430: “For example, translocation experiments for exploring the cultural basis of migratory behaviour, such as those conducted on big horn sheep (*Ovis canadensis*) and moose (*Alces alces*), provide strong evidence for the importance of cultural behaviour for conservation reintroductions [76], (see ESM S4b).” This reference is also now included in Figure 2 (along with other species references; see below).*

Ln 501: Here and in figure 3, the authors need to provide more information about how assessing phylogenetic propensity for social learning/cultural transmission would be accomplished. Even if it is just a sentence or two with some citations. Otherwise, the proposed approach is vague and requires the reader to guess and make leaps of faith as to what the authors are suggesting.

A: We have removed reference to phylogenetic inferences.

Ln 518-521: Here and other areas of the manuscript, particularly the section on resilience, resistance, and recovery, is an opportune place to integrate some discussion of the role of cultural evolution in conservation outcomes. For example, cultural evolution may be a source of rapid adaptation to various aspects of global change.

A: Due to space requirements, Section 3c Resilience, has been moved to the ESM. However, to emphasise this important point, we have included the following text in Section 5 at L483: “Social learning and thus cultural evolution may provide opportunities for adaptive behaviours to spread in response to environmental change[84]. Conversely, social learning may prevent the spread of adaptive behaviour, potentially hindering recovery, if conformity is high or some other mechanism promotes cultural ‘conservatism’ [48]. It may also have a subtle and complex role in resistance to disturbance as the result of knowledgeable elders acting as repositories of social knowledge, as for example in African elephants and killer whales [56,85].”

A: Additional text has been included in Section 5 to bring forward this link, and the link between culture and vital rates, for managers. L493 “Specifically, understanding linkages between culture and vital rates, cultural evolution, and adaption to rapid global change ,will be critical for incorporating culture into management plans”.

Figure 2: In the case of bighorn sheep in North America, the mitigation strategy suggested by the authors is not viable. In Jesmer et al. 2018, bighorn sheep and moose were translocated into areas of

their historic range where the species was previously extirpated. So, there were no knowledgeable individuals to translocate because no individuals had experience on those landscapes. I do not see a way to introduce/translocate knowledgeable individuals in reintroduction efforts. One potential mitigation strategy would be to perform some sort of 'assisted migration' (not in the way the term is typically used) where humans harness emulation or local enhancement to 'seed' knowledge in a subset of individuals (perhaps by training captive reared individuals in the natural setting a la Mueller et al. 2013, Science) about where and when to migrate, then see/hope if that information diffuses through a population.

A: To clarify this point, under the mitigation strategy column for bighorn sheep in Figure 2, we have edited the text from: "Introduce with knowledgeable individuals", to "Potential to harness emulation or local enhancement by intervening to seed knowledge in a sub-set of individuals".

Although space is always limited in print journals, and authors often feel forced to shift as much information as possible to supplemental materials (myself included), I strongly suggest a 7th column of references be added here. The information in the supplement will relatively rarely be read and authors of these works will not get appropriate accreditation. Given most of the studies used in this table are cited in the main text, I do not think it will add too many citations to the author's reference section to add a references column.

A: We very much appreciate this balance and have included one reference per species, located under the species name, as space is tight, and we are unable to accommodate all references from the Supplement here. We have edited the Figure caption to direct readers to the additional references located in ESM table S2.

Figure 3: In figure 3 and throughout the manuscript I suggest spelling out acronyms (e.g., ESU, DIP, CV, CMS, CITES, IPBES, COSEWIC, etc.). These acronyms are not used over and over throughout the manuscript, so spelling them out will not cost much space but will certainly make the manuscript easier to read for non-experts. Also, with regards to using acronyms in figure 3, non-expert readers will have to search through the main text or supplemental materials to decipher the acronyms and interpret the figure.

A: To address these points and increase clarity we have added common names under the silhouettes in Figure 3. We have also spelled out each acronym in Q3.2 and Q3.3 to ensure it is easily understandable to non-experts.

Also, a more minor point regarding figure 3 is that bighorn lambs do not have full-curl horns. Suggest finding a silhouette with no or very small horns or using image editing software to delete the horns on the current silhouette.

A: The horns on the lamb have been removed.

Minor comments:

Ln 163: What are "precautionary principles"? If this is jargon, I suggest defining. If the authors are referring to discussion preceding this paragraph, it remains unclear as to what these principles are.

A: We have included a definition of this term in the Glossary (ESM S2) to ensure clarity for those not familiar with this management approach. ESM S2, Pg 2: “Precautionary principle: At its most basic, the precautionary principle is a principle of public decision making that requires decision makers in cases where there are ‘threats’ of environmental or health harm not to use ‘lack of full scientific certainty’ as a reason for not taking measures to prevent such harm [10].”

Ln 175-179: This is an important message that I think will be appreciated by those interested in studying animal culture in wild populations where such ‘gold standards’ are impractical or impossible to implement. Thank you.

A: Thank you.

Ln 209: Suggest describing what is meant by "viewing culture in isolation" or reword. Unclear what is meant here.

A: This sentence has been reworded for clarity L230 “Culture can be one of many influences that shape behaviour and new modelling approaches now integrate ecological, social and genetic factors into analyses (e.g.,[36]).”

Ln 219: Two complete sentences combined with a comma (i.e., run on sentence).

A: Edited to two sentences (now L239-241).

Ln 389: Typo. Should read “may not yet be”

Ln 389-393: Reads awkwardly, suggest moving “thus providing a fertile area for on-going research” to the end of the sentence.

A: Typo fixed and sentence reworked (L369). “While linkages to vital rates and conservation impacts may not yet be established for many species, we hope the examples (ESM S4a-c, figure 2) will encourage readers to re-examine their data using a cultural lens to investigate whether social learning is important for conserving their focal species.”

Ln 438-441: Suggest moving this sentence to after the sentence on the use of genetic pedigrees. Otherwise, the sentence on biologging interrupts the logical linkage between the first and third sentences about the use of genotypic data.

A: Moved as suggested (see L427-430).

We provide below the tracked-changes manuscript and supplement. Please note that in the supplement (ESM S4(a), p.7) we reference unpublished data for a manuscript which is currently under review. We would be grateful for your advice on how best to refer to this material in the supplement. Thank you for further consideration of our manuscript. If there are any further questions or comments, please do not hesitate to contact us.

Yours sincerely,

Philippa Brakes, Ellen Garland and Emma Carroll, on behalf of co-authors

Appendix B

Response to Referee

Reviewer(s)' Comments to Author:

Referee: 1

Comments to the Author(s)

This is the second time I have reviewed “A deepening understanding of animal culture brings novel perspectives to conservation”. The authors did a good job addressing most of my previous comments and I thank them for taking the time to do so.

1. *Thank you. We endeavoured to address all of your comments.*

However, I still can't help but get the feeling that this paper primarily about animal social learning and culture and the conservation aspect is only considered secondarily. Perhaps I am just not getting it – but I still don't feel like the link to conservation is very strong and I still found that many of the examples are great in the context of culture and social learning, but they are less relevant for the conservation and management of species. For example, that culture is related to survival in bottlenose dolphins (a very common species) is not particularly relevant for the conservation of any other species.

2. *First, the paper provides a strong background on social learning and animal culture, and how it is studied, precisely because it is aimed at an audience that includes many relevant readers who are unlikely to be familiar with the field and the possible linkages to conservation. Our approach has been to highlight (1: Section 3) processes through which social learning and culture influence fitness and population parameters and therefore may be important for conservation and management (i.e., survival, reproduction), and (2: Section 4) areas of research that might be particularly fruitful to examine in the context of culture and conservation (i.e., migration, foraging, communication).*
3. *Second, we use a range of examples to highlight the potential impacts of cultural processes on fitness and demography across a wide range of species and contexts (with the implication that these illustrations of social learning may be far more widespread than the handful of well-studied model systems; e.g., Kasuya 2008). For every case study we have included we provide a clear explanation (or suggestion) about how this might be relevant to conservation. However, we have taken on board the Reviewer's comments about the lack of clarity regarding the link between some of the examples and their relevance to conservation and management. To address this, we have clearly stated how chosen non-threatened species examples highlight important processes that could be shaped by social learning and culture and that could have relevance to other species. For example, conservation outcomes depend on demographic processes. If social learning influences demography then it stands to reason that conservation policy must take it into account. We have highlighted in the text examples of the link between social learning and demography where it is explicitly used in conservation (e.g., killer whale cultural units – see #13).*
4. *Finally, we respectfully disagree with the reviewer's implication that only data from endangered species is relevant to this discussion, for the following reasons:*

- a) *If social learning has fitness implications in one species, it is a reasonable prediction that it may have similar implications in other, similar species (particularly if there is similarity in diet, morphology, social system, etc; Kasuya 2008). Ignoring this possibility simply because data are lacking for the particular species of conservation interest would be negligent.*
- b) *By definition, endangered species are often scarce and therefore elusive and difficult to study. For this reason we have to rely largely on studies of non-endangered model systems to understand fundamental principles of social learning and cultural transmission, and how these might impact fitness and population dynamics. These insights from model systems can then be applied to address conservation problems in other taxa. Relevant examples of this include (a) the golden lion tamarin reintroduction programme, which was developed on the basis of fundamental insights gained from studies of social learning in animals like rats and captive primates (e.g., Galef & Giraldeau 2001, Mineka & Cook 1988)¹; (b) the development of anti-predator and survival training in critically endangered Hawaiian crows, which builds on insights derived largely from work on other, non-endangered bird species (Greggor et al. 2016)²; (c) the whooping crane example which builds on studies of imprinting and social learning of migratory routes in other (non-endangered) bird species. Many successful conservation interventions depend on insights derived from research on other non-endangered species. Southern right whales are discussed in Section 4b of the manuscript, as an example of how data on social learning has influenced the designation of management units in response to the patchy recovery of a species following hunting. However, this is also an example of how management based on data about social learning has resulted in this species moving positively between threat categories in New Zealand, from ‘threatened’ (nationally vulnerable), down to ‘at risk’ (recovering)³. This demonstrates further the value of highlighting examples from species that are recovering.*

5. *Specific changes to the manuscript have been made as detailed below to clarify the general points made above. We have now included clear statements regarding the choice of specific case study species i.e. whether this is to highlight links to conservation and management, or if this is to highlight a cultural process typically illustrated in a species that may not appear on first inspection to be relevant for conservation. For example:*

6. *L115-121 “We provide examples where the linkages between conservation and social learning have been demonstrated for endangered species. However, to further*

¹ Galef Jr, B.G. and Giraldeau, L.A., (2001). Social influences on foraging in vertebrates: causal mechanisms and adaptive functions. *Animal behaviour* 61(1):3-15.

Mineka, S., & Cook, M. (1988). Social learning and the acquisition of snake fear in monkeys. In T. R. Zentall & B. G. Galef, Jr. (Eds.), *Social learning: Psychological and biological perspectives* (p. 51–73). Lawrence Erlbaum Associates, Inc.

² A.L. Greggor, N.S. Clayton, A. Fulford, A. Thornton, (2016). Street smart: Faster approach towards litter in urban areas by highly neophobic corvids and less fearful birds, *Animal Behaviour*, 117:123-133

³ Baker CS, Boren L, Childerhouse S, Constantine R, van Helden A, Lundquist D, Rayment W, Rolfe JR. (2019). Conservation status of New Zealand marine invertebrates, 2019. *New Zeal. Threat Classif.* 29. (doi:10.1080/00288330.2010.495373)

elucidate some of the underlying cultural and demographic processes, we also provide examples from species of lower conservation concern, to assist researchers and practitioners in identifying scenarios where social learning may be important for the conservation of endangered species, or for distinct population segments. ”

7. *L196-199 “Indeed, controlled studies can be vital for informing conservation by shaping our understanding of the fundamental principles of social learning and cultural transmission, and how they interface with demographic processes, e.g., anti-predator and survival training [26].”*
8. *L277-285 “To illustrate the links between these demographic parameters and social learning, we draw on examples from a wide variety of species, of varying threat level. The processes elucidated in these examples have relevance for the management of many species, regardless of their conservation status. Indeed, while some examples in this section may not be of immediate conservation concern, many countries actively manage species and populations to avoid them slipping into such categories; therefore, understanding the influence of culture on demographic processes is highly relevant.”*
9. *L323-336 “Building on innovative research on model organisms [20,22,37], consideration and utilisation of social learning has proved important for increasing survival in managed populations ([2], ESM S4a).”*
10. *L447-450 “These contexts are often the focus of conservation actions. Therefore, our aim is to provide a roadmap to understand the contexts under which social learning may be relevant and to consider ways the field can contribute to promoting conservation outcomes.”*
11. *We also note that many species are managed that are not of conservation concern. The following text has been included to clarify this point. L595-598 “The examples given here are relevant to endangered species, but may also provide insights for those species not currently of conservation concern; managers work to ensure that populations do not decline into threatened status, after all.”*
12. *We have also added the specific and most relevant examples of culture informing conservation management at an international level: CMS Concerted Actions. L133-137 “‘Concerted Actions’, approved by the Parties to the treaty, based on cultural data now inform the conservation management of eastern tropical Pacific sperm whales (*Physeter macrocephalus*) and ‘nut-cracking’ Western chimpanzees (*Pan troglodytes verus*) (ESM S1, S4a, S4c) under CMS.”*
 - a) *Concerted Actions text added to sperm whale example in Section 3b. L393-402 “Foraging variation amongst clans can lead sub-populations to respond differently to environmental change, such as the El Niño oceanographic phenomenon. Noting this differential success between acoustic clans, in 2017 the Parties to CMS agreed a Concerted Action to further explore the implications of the clan structure for the conservation of sperm whales in the eastern tropical Pacific [57]. While the influence of social learning on reproductive success is apparent, it is not yet clear how environmental changes influencing feeding success impact clan survival; such information is essential for understanding population dynamics within clans and across the species.”*
 - b) *Concerted Actions text for chimpanzee example added to Section 4a. L488-495 “For example, multiple lines of evidence have now established nut-*

cracking, a foraging specialisation limited to sub-populations of critically endangered Western chimpanzees, as a socially learnt and culturally transmitted behaviour that may be essential to survival through the dry season when fruit is scarce. Noting this specialisation and the critically endangered status of these sub-populations, in 2020 the Parties to CMS agreed a Concerted Action to further explore the implications of nut-cracking culture for the conservation of this species (ESM S1, S4a).”

Within section 3 ‘Conservation through the lens of social learning and culture’ – I found the link to conservation to be tenuous. The introductory section provides examples are only vaguely linked to potential conservation issues through things like “resource scarcity” – an ecological process that happens in all environments for all species (so this is not special for species at risk). See below for comments on the specific line numbers.

- 13. This is an example of a processes driven link. While resource scarcity can be a general ecological process, the issue we are raising here is that social learning can result in different segments of the population being differentially affected by ecological perturbations, such as prey abundance e.g., if they are reliant on different prey resources. We refer the reviewer to Figure 1, which provides an overview of conservation units (ESUs, DIPs, CVs) and how they are used in current conservation frameworks. We have also added the following example of such a process to Section 3c. L424-432 “For example, killer whales (*Orcinus orca*) can exhibit highly conservative socially learnt prey specialisations to the extent that separate, endangered fish-eating Southern Resident killer whale social units forage on fish (e.g., chinook salmon, *Oncorhynchus tshawytscha*) specific to individual river systems [58]. The population abundance of this social unit has declined along with its preferred prey. This reliance on a single river system and cultural reluctance to switch food sources clearly links the importance of understanding foraging culture with conservation management.”*

As this section progresses through section (a) to section (b) I felt as though I was reading a review on the link between culture and demography, not culture and conservation. Conservation certainly includes aspects of demography and the role of culture and social learning on vital rates is very important, but I still feel like the link to conservation is not as central as it could be. Specifically, I think the implication here is that if culture or social learning influence survival or reproduction than this is good for conservation. Sure, this may be true, but I feel [sic]

- 14. Please see our approach above (e.g., bullet points 2-4). To address this specific comment, we have clarified in the text that we are discussing both conservation AND management (where both the positive and potential negative implications of social learning and culture can play a role). We contend that effective management includes preventing species declining into threatened status. While many bottlenose dolphin populations may not be of immediate conservation concern, some jurisdictions actively manage these populations and monitor the threats that they face. Thus, understanding the implications of cultural processes for some of these*

populations – including the culturally mediated sub-groups that they can form - can be highly relevant to policy.

I do think the golden lion tamarin example is a good one, but I find the bottlenose dolphin example to be a bit more tangential – aren't bottlenose dolphins one of the most common dolphin species in the world? I think the dots need to be better connected from social learning -> survival -> conservation efforts (in the tamarin example, the authors do this, but I think the dolphin example on line 297-301 one falls short).

15. *To complement the tamarin example, we have brought forward another captive reared example, the critically endangered Hawaiian crow, from the ESM to further support Section 3a. L344-354 “In another example, to maximise post-release survival of captive reared critically endangered Hawaiian crows (*Corvus hawaiiensis*), young birds are conditioned to recognise a potential natural predator, the Hawaiian hawk (*Buteo solitarius*), and to exhibit context-appropriate anti-predator behaviour¹. In addition to learning to avoid danger, Hawaiian crows may socially learn key skills required to forage efficiently, communicate in a species-typical manner, and breed successfully ([49], see ESM 4a). These examples illustrate the importance of seeking to maintain individuals as ‘repositories of knowledge’ that may span a number of behavioural contexts and ensuring individuals scheduled for release are behaviourally competent, thus impacting conservation success.”*
16. *We have re-arranged Section 3a (start L322) to highlight examples of endangered species first and follow this with the bottlenose dolphin example. The bottlenose dolphin case study provides insights on how social learning can influence demographic processes relevant to conservation and management for a species managed under a government legislative framework (e.g. the Australian Environment Protection and Biodiversity Conservation Act). Finally, this example on the survival effects in bottlenose dolphins, may be applicable to other endangered dolphin species and threatened bottlenose dolphin populations and thus should be taken into consideration in conservation policy.*

Similarly in section (b), I find the bottlenose dolphin example between lines 321-324 is really interesting and important – but what is the link to conservation? Same thing goes for the sperm whale example between lines 329-333 – what is the link to conservation?

17. *We agree with the reviewer that the second bottlenose dolphin example is interesting but have removed it to accommodate expansion of the sperm whale and elephant examples where clear conservation management linkages are demonstrated.*
18. *We have expanded the sperm whale example into its own paragraph to explicitly demonstrate the link between social learning, acoustic cultural clans and conservation efforts. We note that a CMS Concerted Action plan for management of multiple sub-populations of sperm whales in the eastern tropical Pacific has been developed within CMS based on cultural differences between acoustic clans of sperm whales. (New text provided in #12a).*
19. *The sperm whale case study provides: 1) a clear statement of conservation management of these sub-populations/cultural units through an international body*

- (CMS), and 2) highlights some of the data deficiencies that need to be addressed to better understand the impact of conservation threats and culture on these groupings.
20. We have also clarified the elephant example to emphasise that this information should be taken into account by managers when making any translocation or culling decisions. L377-387. *“At a group scale, the sharing of social information by experienced older African elephant (*Loxodonta africana*) matriarchs increases group survival and reproductive success, by providing information on the level of threat posed by elephants from other social groups and by predators in the wider environment [53]. Management plans should incorporate the understanding that matriarchs act as ‘repositories of knowledge’ and that loss of these individuals (e.g., culling or translocation) can have population-level impacts that persist for decades [54].”*

One general comment I have is that in many places, conservation is focused on habitat protection and less focused on protecting a single species – but this does not really come up except only indirectly in the migration section. What is the relevance of culture or social learning within the context of protecting large swathes of habitats?

21. *This is a fair point, but it relates to species-focused conservation generally, not just the social learning approach we’re taking here. However, an extensive examination of this topic is beyond the scope of the current manuscript, although we highlight the link to habitat and ecosystems based management in figure 3. There’s already an extensive literature on species vs habitat-based conservation, which highlights that conservation efforts for flagship species/species that provide important ecosystem services will benefit habitats as a whole (see Simberloff 1998)⁴.*

Perhaps the authors might consider shifting the focus of the paper away from the extensive series of examples provided in text and using this space to focus more generally on the link between culture/social learning and the variable of interest (i.e. survival, reproduction, units to conserve, foraging, migration, or communication). I think having the examples in Figure 2 is great, but so many of the in text examples don’t have a connection to conservation. I think getting more into the ecological and evolutionary underpinnings of culture and social learning and linking these general ideas to the conservation and management of species or habitats might be a more compelling way to demonstrate the importance of culture/social learning. With this type of narrative the authors could then develop more of a theoretical framework that does not rely on so many disparate empirical examples and they could focus on providing more detail on a smaller number of examples (e.g. the golden lion tamarin example).

22. *Above, we outlined our general approach (bullet points 2-3); to draw on the existing body of literature to highlight (1) processes through which social learning and culture influence fitness and population parameters (i.e., survival, reproduction) and (2) areas of research that might be particularly fruitful to examine in the context of culture and conservation (i.e., migration, foraging, communication). This seems to be along the lines of the ‘links’ based approach that the reviewer has suggested, and*

⁴ Simberloff, D., 1998. Flagships, umbrellas, and keystones: is single-species management passé in the landscape era?. *Biological conservation* 83(3): 247-257.

that revising the text as outlined above has shown the parallels between our and the reviewer's proposed approach.

- 23. We have refocused our examples in the text (e.g., sperm whales, elephants, bottlenose dolphins, killer whales) to draw the linkages as requested. We have included other clear examples – Hawaiian crows and also West African chimpanzees – where there is a critical conservation concern, and social learning and culture are integrated into management.*
- 24. While we appreciate the suggestion of the development of a theoretical framework, this paper is aimed at utilising the existing theory and burgeoning empirical discoveries in the field of animal social learning and culture to develop a practical conceptual framework. As outlined in Section 5, figure 3 provides a practical way to assess and perhaps re-evaluate existing data for links between social learning, culture, and facets of conservation biology. The authors of the paper come from a wide range of fields, and have converged on this conceptual framework as a practical guide.*

I hope my review has been helpful and apologize if it is disappointing or if I am wrong in my assessment.

- 25. We have endeavoured to reframe our manuscript following your suggestions.*